# Development of an α-synuclein knockdown peptide and evaluation of its efficacy in Parkinson's disease models

Jack Wuyang Jin[1,8], Xuelai Fan[1,8], Esther del Cid-Pellitero[2,8], Xing-Xing Liu[2], Limin Zhou[3,4,5,6], Chunfang Dai[3,4,5,6], Ebrima Gibbs[1], Wenting He[3,4,5,6], Hongjie Li[3,4,5,6], Xiaobin Wu[3,4,5,6], Austin Hill[7], Blair R. Leavitt [7], Neil Cashman [1], Lidong Liu[1], Jie Lu[1], Thomas M. Durcan[2], Zhifang Dong [3,4,5,6✉], Edward A. Fon [2✉] & Yu Tian Wang [1✉]

Convincing evidence supports the premise that reducing α-synuclein levels may be an effective therapy for Parkinson's disease (PD); however, there has been lack of a clinically applicable α-synuclein reducing therapeutic strategy. This study was undertaken to develop a blood-brain barrier and plasma membrane-permeable α-synuclein knockdown peptide, Tat-βsyn-degron, that may have therapeutic potential. The peptide effectively reduced the level of α-synuclein via proteasomal degradation both in cell cultures and in animals. Tat-βsyn-degron decreased α-synuclein aggregates and microglial activation in an α-synuclein pre-formed fibril model of spreading synucleinopathy in transgenic mice overexpressing human A53T α-synuclein. Moreover, Tat-βsyn-degron reduced α-synuclein levels and significantly decreased the parkinsonian toxin-induced neuronal damage and motor impairment in a mouse toxicity model of PD. These results show the promising efficacy of Tat-βsyn-degron in two different animal models of PD and suggest its potential use as an effective PD therapeutic that directly targets the disease-causing process.

[1] The Djavad Mowafaghian Centre for Brain Health and Department of Medicine, University of British Columbia, Vancouver, BC, Canada. [2] McGill Parkinson Program, Neurodegenerative Diseases Group, Department of Neurology and Neurosurgery, Montreal Neurological Institute, McGill University, Montreal, QC, Canada. [3] Ministry of Education Key Laboratory of Child Development and Disorders, Children's Hospital of Chongqing Medical University, Chongqing, China. [4] National Clinical Research Center for Child Health and Disorders, Children's Hospital of Chongqing Medical University, Chongqing, China. [5] China International Science and Technology Cooperation Base of Child Development and Critical Disorders, Children's Hospital of Chongqing Medical University, Chongqing, China. [6] Chongqing Key Laboratory of Translational Medical Research in Cognitive Development and Learning and Memory Disorders, Children's Hospital of Chongqing Medical University, Chongqing, China. [7] Department of Medical Genetics, Centre for Molecular Medicine and Therapeutics, University of British Columbia, Vancouver, BC, Canada. [8] These authors contributed equally: Jack Wuyang Jin, Xuelai Fan, Esther del Cid-Pellitero. ✉email: zfdong@aliyun.com; ted.fon@mcgill.ca; ytwang@brain.ubc.ca

Parkinson's disease (PD) is a major neurodegenerative disorder. It currently lacks a clinically relevant treatment that can directly target the disease-causing processes. Current clinical approaches, like deep brain stimulation and pharmacological treatments with levodopa and dopamine agonists, only relieve symptoms. The efficacy of these treatments is largely limited by their undesirable complications and side effects[1,2]. Accumulating evidence supports the theory that the nigral dopaminergic neuronal loss, along with main symptoms in PD, is at least partly associated with α-synuclein protein aggregation in the cytoplasm of these neurons[3,4]. Indeed, knockdown of α-synuclein using genetic manipulations, such as antisense oligonucleotide and small interfering RNA (siRNA), has shown protection of dopaminergic neurons in various models of PD[5–9]. The clinical translation of these manipulations into an efficient PD therapy has however been hindered, in part, due to their limited ability to cross the blood–brain barrier (BBB) and the plasma membrane of neurons in the affected areas of the brain. Although several recent studies suggest that this may be partially improved by coupling siRNA with a brain delivery vehicle[10–12], there remains an urgent need for developing new α-synuclein knockdown strategies that are more practical for therapeutic use in human patients. Here we report the development of a short, BBB and plasma membrane-permeant synthetic peptide that can rapidly reduce endogenous α-synuclein via proteasomal degradation. Using both in vitro and in vivo models of PD, we provide proof-of-principle evidence for using this small α-synuclein knockdown peptide as a potential PD therapy.

## Results

### Development of the α-synuclein knockdown peptide, Tat-βsyn-degron.

In order to develop a more clinically applicable α-synuclein knockdown strategy as a potential PD therapy, we modified a peptide-based method that we previously developed to rapidly and reversibly decrease the levels of endogenous proteins by directing them for degradation[13]. Proteins can be degraded by targeting them for either lysosomal or proteasomal degradation in the cell, but depending on pathological conditions, lysosomes, proteasomes, or both can become compromised in PD[14–19]. In previous work, we used a chaperone-mediated autophagy targeting motif to direct α-synuclein into the lysosomes for degradation[13]. Here we test if we can also use a proteasomal targeting signal, instead of a lysosomal targeting signal, for targeting α-synuclein to the proteasomes for degradation. As shown in Fig. 1a, the proposed α-synuclein targeting peptide (Tat-βsyn-degron) is composed of three domains: (1) the plasma membrane transduction domain Tat, which is capable of delivering peptides across both the BBB and the plasma membrane of neurons following a systemic administration in freely moving animals and humans[13,20], (2) the α-synuclein-binding domain βsyn (β-synuclein), which is derived from amino acids 36–45 of βsyn and has been shown to specifically bind to monomeric α-synuclein with high affinity[21], and (3) the proteasomal targeting domain degron, a peptide signal that has been shown to be sufficient to direct its tagged proteins to proteasomes for degradation[22].

As illustrated in Fig. 1b, to determine the efficacy and specificity of βsyn (36–45 of βsyn) as the binding domain of our α-synuclein targeting peptide to induce degradation of α-synuclein, we first constructed two FLAG-tagged targeting peptide mini-genes (FLAG-βsynN-degron and FLAG-βsyn-degron) encoding either natural or reverse amino acid sequences between 36 and 45 of βsyn (both of which have been shown to have a similar binding affinity for monomeric α-synuclein in an in vitro binding assay in a previous study[21]), along with the degron targeting signal. A control mini-gene encoding FLAG-

βsyn without degron was also constructed. Human embryonic kidney 293 (HEK 293) cells were co-transfected with a human α-synuclein plasmid and one of these mini-genes. As predicted, co-transfection of FLAG-βsynN-degron or FLAG-βsyn-degron (Fig. 1c, d), but not the control FLAG-βsyn peptide (Fig. 1e), resulted in a robust reduction in recombinant α-synuclein levels in a dose-dependent manner. This suggests that the targeting peptides, when co-expressed with α-synuclein, are sufficient to bind to α-synuclein and target it for proteasomal degradation. Since FLAG-βsyn-degron appears to have a better α-synuclein knockdown efficacy in comparison with FLAG-βsynN-degron (Fig. 1c, d), possibly due to its enhanced stability[23], we chose to use the βsyn-degron as the α-synuclein targeting peptide to knock down endogenous α-synuclein for all the following experiments. The FLAG-βsyn-degron induced knockdown is target-specific, because this knockdown was not associated with a detectable change in β-actin levels (Fig. 1c, d) and, more importantly, it was selective to α-synuclein, but not β- or γ-synuclein, which are two other members of the synuclein protein family (Fig. 1f).

We next examined the efficacy of the peptide to decrease endogenous α-synuclein in neurons in situ, using chemically synthesized, membrane-permeant α-synuclein targeting peptide, Tat-βsyn-degron, along with the control Tat-βsyn peptide, as illustrated in Fig. 1a. We first confirmed using biacore peptide–protein binding assays that similar to Tat-βsynN-degron, Tat-βsyn-degron specifically bound to purified recombinant α-synuclein (Fig. 2c–e), while Tat control peptide (Fig. 2a) and HEPES buffered saline (HBS) blank buffer control (Fig. 2b) showed little binding to purified recombinant α-synuclein. As shown in Fig. 3a, bath application of Tat-βsyn-degron, but not the control Tat-βsyn, for 24 h produced a dose-dependent reduction of endogenous α-synuclein in cortical neuron cultures. The Tat-βsyn-degron-induced knockdown is mediated by proteasomal degradation, as it could be fully prevented by the presence of the proteasomal inhibitor MG132 (10 μM; 24 h; Fig. 3a). In addition, the knockdown is time-dependent, and the α-synuclein level remained low at 24 h (Fig. 3b). The peptide-mediated knockdown is specific to α-synuclein, as it did not affect the levels of several other neuronal proteins surveyed in the same treated cultures, including transmembrane protein GABA$_A$ (γ-aminobutyric acid type A) receptor β2/3 subunit, intracellular protein HSP90, and 14-3-3, a known α-synuclein-binding protein (Fig. 3c–e). Interestingly, both Tat-βsyn and Tat-βsyn-degron peptides induced similar and dose-dependent increases of lactate dehydrogenase release in primary cortical neurons (Supplementary Fig. 1), possibly due to the pore-forming property of the Tat peptide on the cell plasma membrane[24].

To demonstrate that Tat-βsyn-degron peptide can decrease α-synuclein levels in vivo, we used M83 transgenic mice that overexpress mutant human A53T α-synuclein[25]. M83 mice, injected intraperitoneally (i.p.) with 40 mg/kg Tat-βsyn-degron peptide, were sacrificed at time points of 6, 12, 24, and 48 h, and α-synuclein levels in the brain were measured by enzyme-linked immunosorbent assay (ELISA). The Tat-βsyn-degron peptide led to a reduction of α-synuclein levels at both 12 and 24 h, but not at 48 h, indicating that the effect of the Tat-βsyn-degron peptide in vivo is transient and peaks around 24 h in this mouse line (Fig. 3f).

### Therapeutic efficacy of the Tat-βsyn-degron peptide in an in vitro model of PD.

Having confirmed the efficacy and specificity of the Tat-βsyn-degron peptide in knocking down endogenous α-synuclein levels in neurons and in vivo, we examined the therapeutic potential of the Tat-βsyn-degron peptide by examining its ability to protect dopaminergic neurons against 1-methyl-4-phenylpyridinium-positive (MPP+) toxicity using a

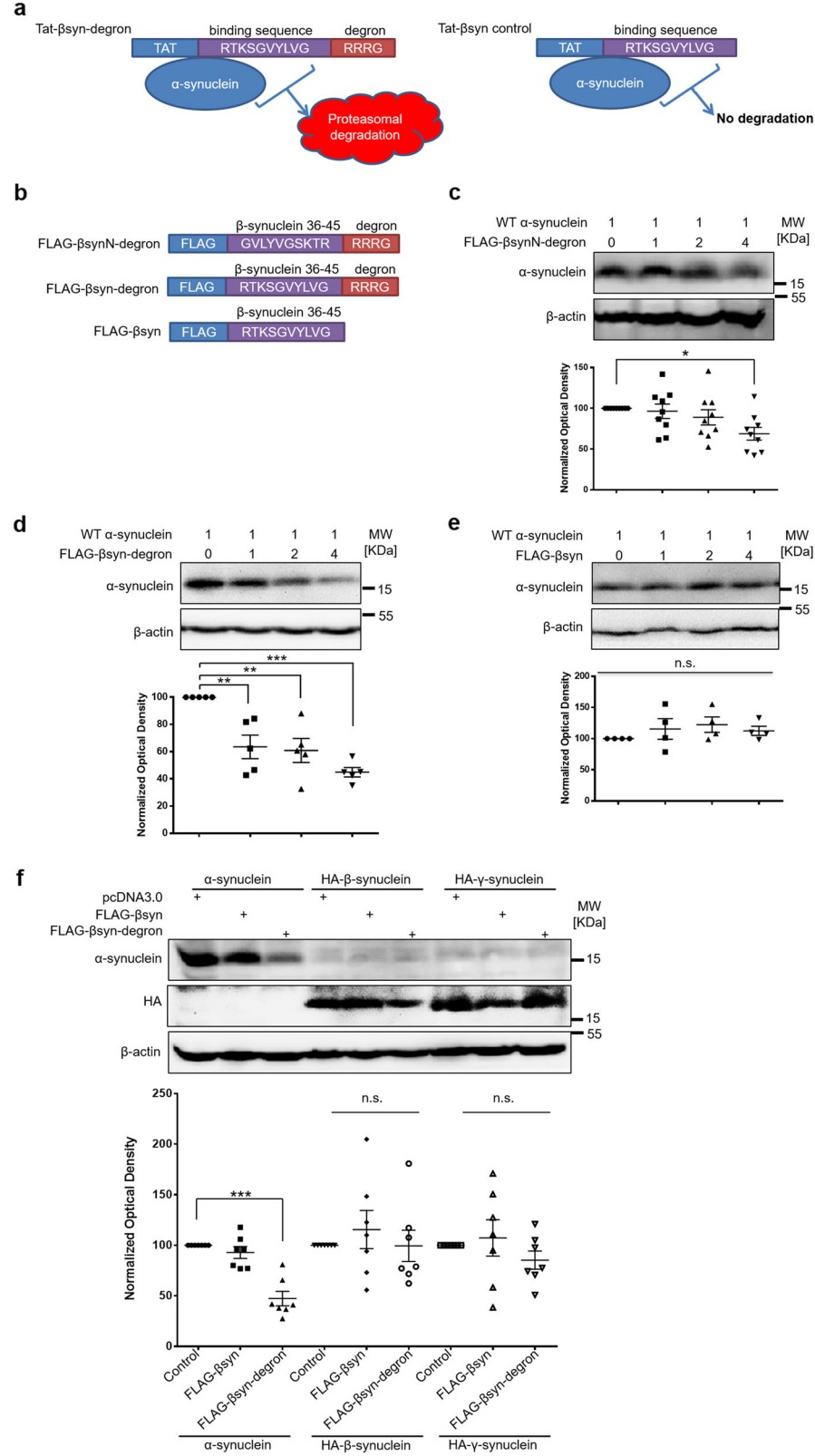

well-characterized in vitro model of PD[26]. As shown in Fig. 4, MPP+ treatment (20 µM; 48 h) induced the dramatic death of dopaminergic neurons in rat primary cultures of the ventral midbrain. This was demonstrated by the significant decrease in the level of tyrosine hydroxylase (TH), a dopaminergic neuronal marker protein (Fig. 4a, c) and in the numbers of TH-positive neurons (Fig. 4d, e). Bath application of the Tat-βsyn-degron peptide (25 µM; 48 h), but not the control Tat-βsyn peptide (25 µM; 48 h), induced a robust reduction in endogenous α-synuclein protein levels (Fig. 4a, b). Importantly, the reduction of α-synuclein almost fully protected dopaminergic neurons from MPP+-induced neurotoxicity, as shown by the rescue of TH protein level (Fig. 4a, c) and TH-positive neurons (Fig. 4d, e) in the culture dishes.

**Fig. 1 The design of α-synuclein knockdown mini-genes and peptides and demonstration of knockdown efficacy in HEK 293 cells. a** Schematic illustration of the Tat-βsyn-degron peptide design. The α-synuclein targeting peptide Tat-βsyn-degron has three domains: (1) the Tat transduction domain that enables the peptide to penetrate cell membranes, (2) the α-synuclein-binding domain derived from β-synuclein, and (3) the degron sequence that targets the Tat-βsyn-degron and α-synuclein complex to the proteasome for degradation. In contrast, the Tat-βsyn control peptide lacks the proteasomal targeting signal, and hence while it can bind to α-synuclein, it cannot direct the complex to the proteasome for degradation. **b** Schematic illustration of the mini-gene constructs encoding FLAG-βsynN-degron (βsynN: natural amino acid sequence between 36 and 45 of β-synuclein), FLAG-βsyn-degron (βsyn: reversed β-synuclein 36–45) or FLAG-βsyn. **c–e** Immunoblots sequentially probing for α-synuclein and β-actin (as loading and specificity controls) showing that expression of FLAG-βsynN-degron (**c**; $N = 9$; $F_{(3,32)} = 3.43$; $P < 0.05$) or FLAG-βsyn-degron (**d**; $N = 5$; $F_{(3,16)} = 13.18$; $P < 0.001$), but not control FLAG-βsyn (**e**; $N = 4$; $F_{(3,12)} = 0.73$; $P = 0.55$), induced a dose-dependent reduction in the levels of human α-synuclein co-expressed in HEK 293 cells. Note that FLAG-βsyn-degron appears to have a better efficacy in reducing α-synuclein. Transfection ratios of the plasmids are shown on the top. **f** FLAG-βsyn-degron-mediated knockdown is α-synuclein specific. Immunoblots sequentially probing for synuclein and β-actin showing that when co-transfected at 4:1 ratio in HEK 293 cells, FLAG-βsyn-degron, not FLAG-βsyn, specifically reduces the level of α-synuclein ($N = 7$; $F_{(2,18)} = 28.51$; $P < 0.001$), but not the levels of HA-β-synuclein ($N = 7$; $F_{(2,18)} = 0.42$; $P = 0.66$) or HA-γ-synuclein ($N = 7$; $F_{(2,18)} = 0.93$, $P = 0.41$). Data are presented as mean ± SEM. The statistical difference between groups was determined by one-way ANOVA, followed by Bonferroni post hoc test. $*P \leq 0.05$, $**P \leq 0.01$, and $***P \leq 0.001$ compared with the control. n.s. denotes not significant. Both solid and open circles, squares and triangles represent individual data points in each group.

**Therapeutic efficacy of the Tat-βsyn-degron peptide in a mouse model of spreading synucleinopathy.** We next tested the therapeutic efficacy of the Tat-βsyn-degron peptide in an animal model of PD. Accumulating evidence supports the role of prion-like propagation of α-synuclein in the pathogenesis of PD[27,28]. To study whether the Tat-βsyn-degron peptide-induced knockdown can reduce the propagation of α-synuclein in the brain, we established a mouse model of spreading synucleinopathy as described previously[29]. M83 mice were injected intracerebrally (i.c.) into the right dorsal striatum with either 12.5 μg of α-synuclein preformed fibrils (PFFs; Supplementary Fig. 2a, b) or phosphate-buffered saline (PBS). Starting at 3 days prior to the PFF injection, mice were treated daily for 12 days with either Tat-βsyn or Tat-βsyn-degron peptide (40 mg/kg; i.p.) and once every other day for the subsequent 8 days (20 days in total, Fig. 5a). At the end of 3 months, coronal brain sections of the M83 mice were prepared and stained with antibodies against serine 129-phosphorylated α-synuclein (pS129syn), a marker of pathogenic synuclein aggregates[30] and Iba-1, a calcium-binding protein specifically expressed in macrophage and microglia as a marker of neuroinflammation[31]. Although we did not observe obvious PD-related behavioral phenotypes in either group of mice when behavioral tests were performed 45 days and 3 months after PFF injection (Supplementary Fig. 3), PFF, not PBS, inoculated animals exhibited some characteristic phenotypes of PD pathology, including increased α-synuclein aggregation and inflammation in defined regions of the brain, including the substantia nigra pars compacta and the pons (Fig. 5b–m). Mice injected with α-synuclein PFFs exhibited increased intensity of pS129syn staining (Fig. 5b–e). In these PFF-injected mice, pS129syn-immunoreactive profiles were found in cell somas, dendritic branches, and axons (Fig. 5c, e). In comparison with the Tat-βsyn group, pS129syn staining was significantly reduced in both substantia nigra pars compacta (Fig. 5f) and pons (Fig. 5g) of the Tat-βsyn-degron-treated group, indicating that the Tat-βsyn-degron peptide reduced propagation and seeding of α-synuclein aggregates in the brains of these mice.

The staining for Iba-1 could be detected throughout the brain in all mice (Fig. 5h–k). We found that in comparison with the Tat-βsyn-treated group, the administration of Tat-βsyn-degron, while having no observable effect in the substantia nigra pars compacta (Fig. 5l), significantly reduced the level of Iba-1 staining in the pons region (Fig. 5m). These results suggest that i.c. inoculation of α-synuclein PFFs produces inflammatory responses and that the inflammatory responses can be reduced in a region-specific manner by knocking down α-synuclein with the Tat-βsyn-degron peptide.

**Therapeutic efficacy of the Tat-βsyn-degron peptide in the MPTP mouse model of PD.** While the aforementioned PFF inoculation PD model demonstrates the effectiveness of the Tat-βsyn-degron peptide in reducing PFF-induced PD pathology, the lack of behavioral alterations, at the time points tested, precluded us from directly evaluating the efficacy of peptide-induced α-synuclein knockdown in reducing PD behavioral phenotypes. We therefore examined the therapeutic efficacy of the Tat-βsyn-degron peptide in a pathologically and behaviorally well-characterized mouse dopaminergic neuron toxicity model of PD[32]. C57BL/6 mice were i.p. injected with 30 mg/kg parkinsonian toxin 1-methyl-4-phenyl-1,2,3,6-tetrahydropyridine (MPTP) (or saline as control), once per day for 5 consecutive days, to induce dopaminergic neuron damage. The effects of MPTP administration on mouse rotarod performance and damage to dopaminergic neurons were then analyzed 1 week after the last injection of MPTP. To determine the effect of α-synuclein knockdown in protecting dopaminergic neurons against MPTP, the Tat-βsyn-degron peptide or its control Tat-βsyn (6 μmol/kg; i.p.) was used in some animals twice a day for 12 days, beginning the first day of MPTP injection. As shown in Fig. 6, the Tat-βsyn-degron peptide, but not Tat-βsyn peptide, induced significant α-synuclein degradation in both substantia nigra-containing ventral midbrain (Fig. 6a, b) and striatum (Fig. 6d, e). The striatum is a region that receives dopaminergic neuronal projections from the substantia nigra and is also deeply affected in PD[33]. Consistent with the specific neurotoxic effects of MPTP on dopaminergic neurons, in mice receiving MPTP injection alone there was a significant loss of protein TH in both dopaminergic neurons in the substantia nigra-containing ventral midbrain (Fig. 6a, c) and dopaminergic neuronal terminals in the striatum (Fig. 6d, f). This was demonstrated by quantitative immunoblotting analysis of TH protein levels. As expected, the specific knockdown of α-synuclein by Tat-βsyn-degron protected against MPTP-induced dopaminergic neuronal injury (Fig. 6c, f).

The neuroprotective effects of Tat-βsyn-degron peptide were further supported by immunohistochemical analysis. In mice receiving only MPTP injections, there was a significant loss of TH-positive neurons in the substantia nigra pars compacta as estimated by blinded neuron counting from bregma −2.92 to −3.64 mm (Fig. 7a, b) and TH-positive neuronal terminals in the striatum as quantified with densitometric analysis (Fig. 7c, d). As expected, the MPTP-induced dopaminergic neuronal damage was largely protected by the Tat-βsyn-degron peptide, but not the Tat-βsyn peptide (Fig. 7a–d).

The motor function of these mice was tested by a rotarod test using a protocol modified from a previous study[34]. Consistent

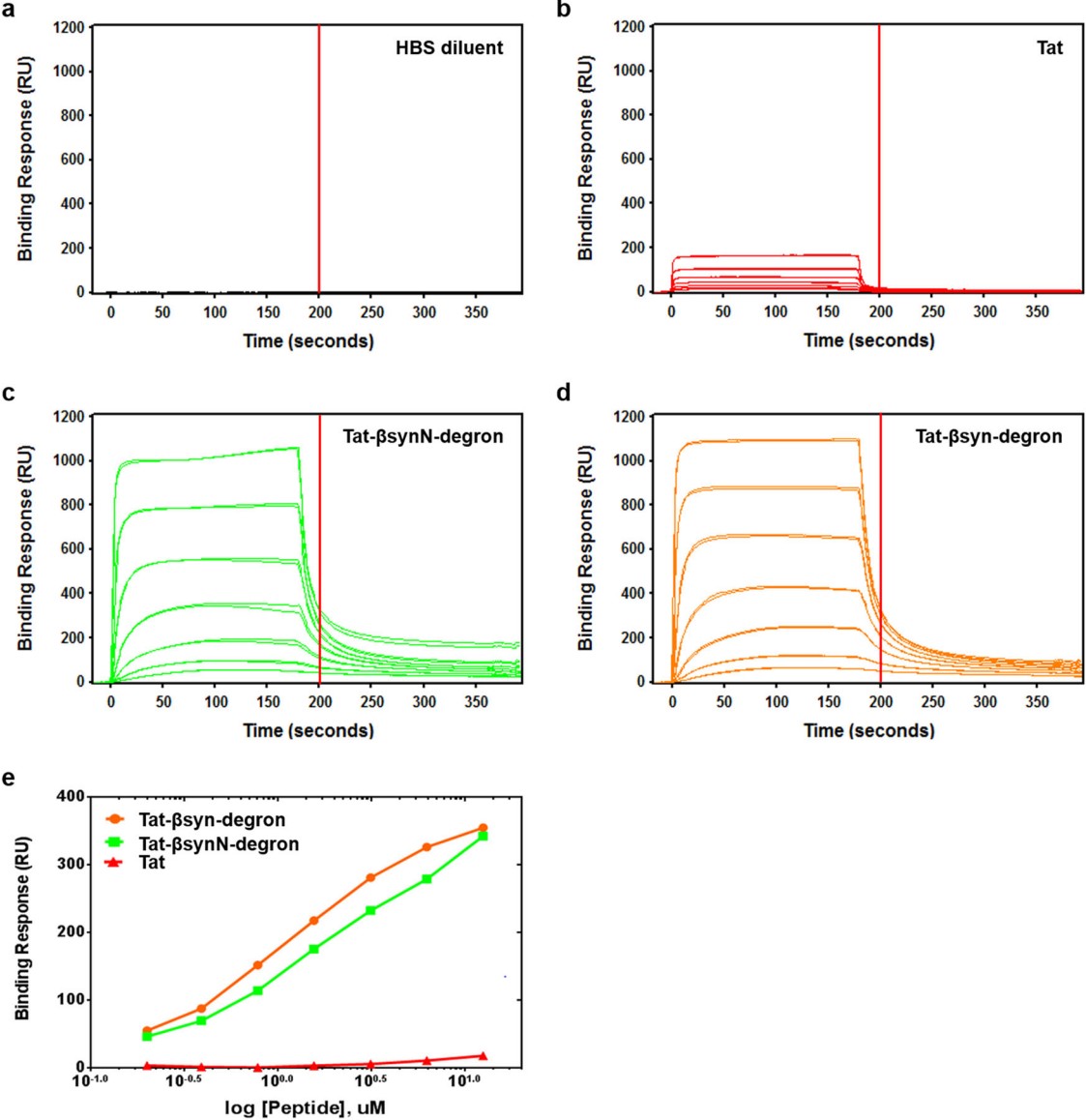

**Fig. 2 Biacore peptide–protein binding assays. a–d** Representative sensorgrams demonstrating the binding responses of the synthetic Tat peptide (**b**), Tat-βsynN-degron peptide (**c**), Tat-βsyn-degron peptide (**d**), or HBS blank buffer control (**a**) to α-synuclein. Two-fold serial dilutions (0.20, 0.39, 0.78, 1.56, 3.13, 6.25, 12.50 μM) of peptides, in duplicate, were sequentially injected over immobilized purified recombinant human α-synuclein for 3 min, followed by a dissociation phase during which HBS buffer was flowed over the surface. Sensorgrams, depicting binding responses over time, were double-referenced by subtracting out the binding on the reference surface and the response from the HBS blank buffer control. Peptide–α-synuclein-binding response report points were collected 20 s into the dissociation phase at time 200 s (as indicated by the vertical lines in the figures), to exclude bulk refractive index changes and nonspecific binding. **e** Graphing of peptide–α-synuclein-binding response vs. peptide concentration showing that synthetic Tat-βsynN-degron and Tat-βsyn-degron peptides displayed robust and similar binding with α-synuclein in a dose-dependent manner (0.20, 0.39, 0.78, 1.56, 3.13, 6.25, 12.50 μM), while the control Tat peptide displayed little binding with α-synuclein.

with the dramatic effects of α-synuclein knockdown and its protection of dopaminergic neurons from MPTP-induced neurotoxicity, the Tat-βsyn-degron peptide also significantly rescued the MPTP-induced motor deficits in a rotarod behavioral test (Fig. 7e). Interestingly, treatment with Tat-βsyn, while producing a small, nonsignificant decrease in MPTP-induced neuronal damage (Fig. 6 and Fig. 7a–d), also reduced motor deficits in mice (Fig. 7e). This effect may be, in part, due to the possibility that Tat-βsyn can function as an interference peptide to inhibit α-synuclein oligmerization[21]. Consistent with this conjecture, homology alignment analysis shows that the amino acids 36–45 of βsyn is quite similar to a sequence found

in the N-terminal lipid-binding domain of α-synuclein[35] (Supplementary Fig. 4), a domain that may be involved in the self-oligomerization among α-synucleins[36,37]. As shown in Supplementary Fig. 5, the Tat-βsyn-degron peptide (6 μmol/kg, i.p.) was also able to similarly and significantly reduce the α-synuclein expression in the kidney, the spleen, the ventral midbrain, and the striatum 6 h after peptide injection (Supplementary Fig. 5). These results indicate that the knock-down of endogenous α-synuclein by the Tat-βsyn-degron peptide is not limited to the central nervous system. The contribution of the peripheral reduction of α-synuclein remains to be determined.

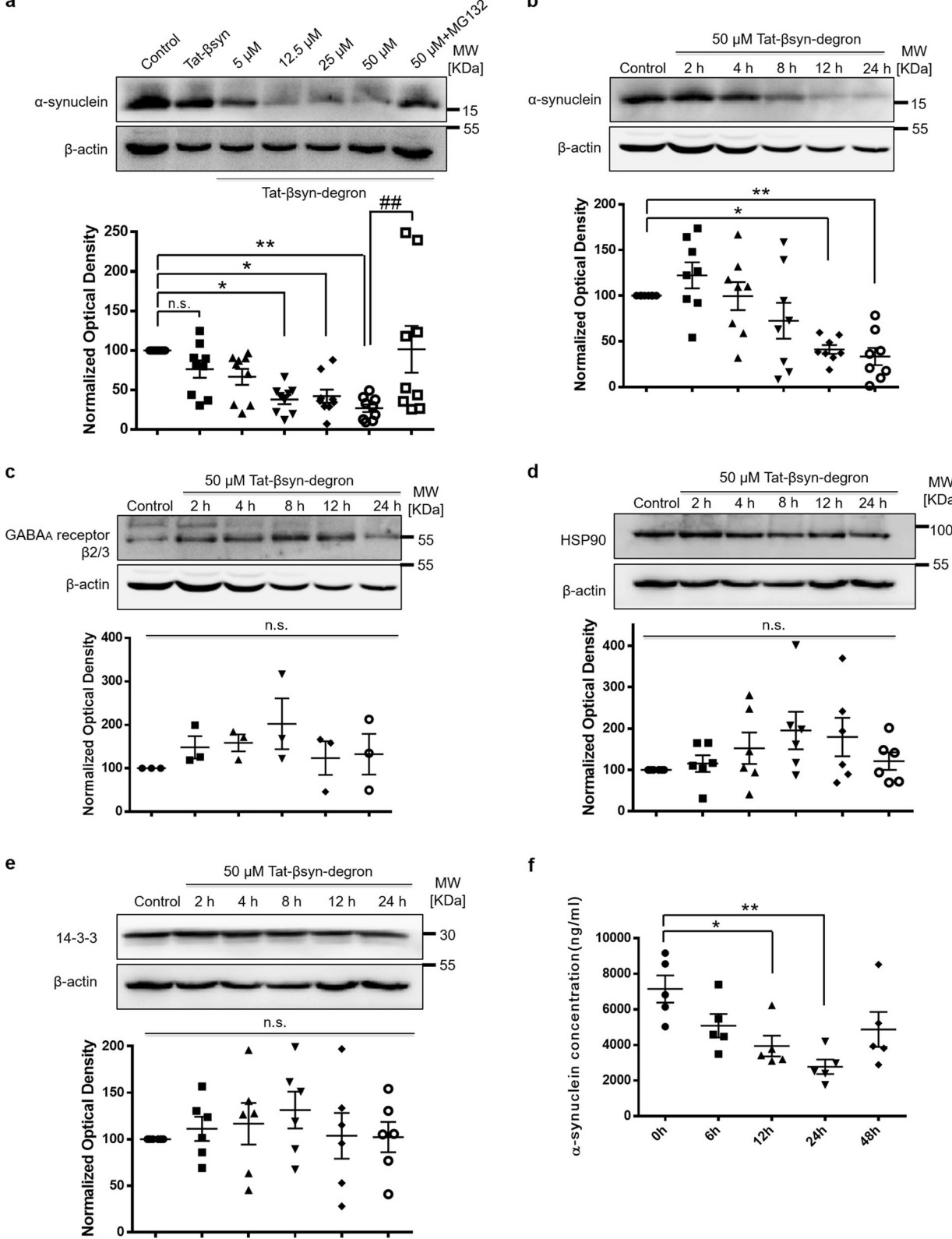

## Discussion

In this study, we provide important evidence supporting that the Tat-βsyn-degron peptide, by rapidly and reversibly reducing the level of α-synuclein, may have therapeutic potential as a clinically applicable treatment for PD. We first demonstrated that the Tat-βsyn-degron peptide can specifically reduce the level of α-synuclein both in vitro and in vivo. We then showed that the peptide-induced α-synuclein knockdown is associated with protection of dopaminergic neurons against toxin-induced damage in a culture model of PD. Most importantly, we were able to demonstrate the therapeutic potential of systemic application of the Tat-βsyn-degron peptide as an effective PD treatment in two well-characterized animal models of PD. Our study not only validates α-synuclein as a therapeutic target upon which new

**Fig. 3 Tat-βsyn-degron peptide dose- and time-dependently knocks down α-synuclein and does not significantly affect the levels of several other cellular proteins. a** Immunoblots showing that bath application of the membrane-permeant synthetic peptide Tat-βsyn-degron at various concentrations for 24 h induced a robust reduction of endogenous α-synuclein protein levels in a dose-dependent manner (5–50 μM; $N = 9$; $F(6,56) = 5.15$; $P < 0.001$); this was prevented in the presence of the proteasome inhibitor MG132 (Tukey's HSD post hoc test: 50 μM Tat-βsyn-degron + 10 μM MG132; ##$P < 0.01$, compared with the 50 μM Tat-βsyn-degron group; $N = 9$). In contrast, bath application of the control Tat-βsyn had no effect (50 μM, 24 h; Tukey's HSD post hoc test: $P = 0.87$ compared with the control; $N = 9$). **b** Bath application of Tat-βsyn-degron (50 μM) induced a time-dependent knockdown of endogenous α-synuclein in primary cortical cultures ($N = 8$; $F(5,42) = 8.06$; $P < 0.001$). **c–e** Tat-βsyn-degron at a high dos**e** (50 μM; 24 h) did not affect the expression levels of several unrelated proteins in cortical cultures. These proteins include transmembrane protein β2/3 subunit of the GABA$_A$ receptor (**c**; $N = 3$; $F(5,12) = 0.90$; $P = 0.51$), cytosol chaperone protein HSP90 (**d**; $N = 6$; $F(5,30) = 1.33$; $P = 0.28$), and 14-3-3, a known α-synuclein-binding protein (**e**; $N = 6$; $F(5,30) = 0.43$; $P = 0.82$). **f** ELISA analysis showing decreased α-synuclein levels at 12 and 24 h in the brain of M83 transgenic mice after an intraperitoneal injection of Tat-βsyn-degron ($N = 5$ per group, $F(4,20) = 5.24$, $P < 0.01$). Data are presented as mean ± SEM. The statistical difference between groups was determined by one-way ANOVA, followed by Tukey's HSD post hoc test. *$P \leq 0.05$ and **$P \leq 0.01$ compared with the control. n.s. denotes not significant. Both solid and open circles, squares and triangles represent individual data points in each group.

disease-modifying therapeutics can be developed for treating PD[38] but also provides the proof-of-concept evidence that the Tat-βsyn-degron peptide developed in this study may represent one of such disease-modifying PD therapeutics.

Our α-synuclein knockdown peptide (Tat-βsyn-degron) is innovative in several aspects: first, by knocking down α-synuclein, one of the major disease-causing molecules, the peptide directly targets one of the disease-causing processes, and can be expected to stop or slow down the progression of the disease. This approach is a stark contrast to most of the PD therapeutic strategies currently used in the clinic. Since deep brain stimulation and currently available pharmacological treatments do not directly target the disease-causing processes, they are, at best, symptom-relieving and cannot stop or slow down the progression of the disease[1,2]. Second, our Tat-peptide-mediated knockdown method also has clear advantages over other protein knockdown technologies, such as antisense oligonucleotide and siRNA. For instance, although siRNA-mediated knockdown of α-synuclein has also been proven to be effective in various models of PD[5,7–9], the clinical applications are hindered by their inability to cross the BBB and the plasma membrane of neurons. The delivery of siRNAs to the brain is mainly accomplished by an invasive i.c. injection or viral infection, which may not be clinically practical for therapeutic use in human patients. Several recent studies suggested that delivery of siRNA to the brain by a noninvasive systemic injection may be achieved by coupling siRNA with brain delivery vehicles such as RVG-9R peptide or RVG-9R peptide-coated liposome[10–12]. These techniques either are restricted to the acetylcholine receptor-expressing neurons in the brain or remain technically challenging. Meanwhile, an antisense oligonucleotide for Huntington's disease has recently entered clinical trials, but it was delivered into patients by intrathecal injection, a complicated procedure that can easily cause discomforts and complications[39]. However, using Tat-mediated protein transduction mechanism, our peptide-based method is much simpler and more effectively delivered into neurons in the brain following a noninvasive systemic administration. The effectiveness was clearly demonstrated by the high efficacy of the peptide in knocking down α-synuclein in the brain and the robust neuroprotective efficacy in two different animal models of PD (Figs. 5, 6, and 7). In addition, the peptide-mediated knockdown has a clear temporal advantage over antisense or siRNA-mediated knockdown. α-synuclein is a very stable protein with a long half-life[40] and it may take a few weeks for antisense or siRNA to induce a significant reduction of endogenous α-synuclein protein level in the brain[7], whereas by hijacking the endogenous proteasomal degradation system in the cell, the Tat-βsyn-degron peptide produced a rapid and robust degradation of α-synuclein protein within a few hours (Fig. 3b, f).

We found that Tat-βsyn-degron peptide reduced α-synuclein levels in a transgenic mouse model (M83)-overexpressing human mutant A53T α-synuclein (Fig. 3f). Using PFF injections into the striatum of this mouse line, we also showed that the Tat-βsyn-degron peptide can reduce the cell-to-cell propagation of pathology, as manifested by a reduction in pS129syn staining and Iba-1 microglial neuroinflammatory infiltration, two well-established hallmarks of synucleinopathy[41,42]. This PFF injection mouse model is increasingly recognized as a robust system reconstituting the prion-like propagation hypothesis of pathogenesis, not only in synucleinopathies such as PD, dementia with Lewy bodies, and multiple system atrophy[43–45], but also in several other neurodegenerative diseases such as Alzheimer's disease and frontotemporal dementia, which involve distinct misfolded proteins such as Aβ and Tau[46,47].

More importantly, the capacity of the Tat-βsyn-degron peptide to show efficacy was further demonstrated in a second animal model of PD (MPTP), further increasing the confidence in our approach. Although α-synuclein aggregation, a pathological hallmark of PD, is not commonly observed in the MPTP mouse model, previous studies had reported that chronic MPTP infusions can induce α-synuclein oligomerization in mice[48–50]. Evidence also suggests that it may be such oligomers that produce most pathological actions of α-synuclein, including inducing neuronal death and motor dysfunction[48–50]. Meanwhile, α-synuclein-null mice display striking resistance to MPTP-induced degeneration of dopaminergic neurons and dopamine release[51]. Given our results indicating that Tat-βsyn can reduce motor deficits without lowering α-synuclein levels (Figs. 6 and 7), we propose that Tat-βsyn, by inhibiting α-synuclein oligomerization[21] (without affecting the amount of α-synuclein), may also be effective in reducing PD-associated motor deficits, as long as a sufficient number of dopaminergic neurons remain functional. This effect, however, may not be long-lasting as it appears insufficient in stopping the PD-associated degeneration of dopaminergic neurons. In contrast, Tat-βsyn-degron has an obvious advantage since it not only rescues the motor deficits but also increases the survival of dopaminergic neurons. This is not surprising given that in comparison with Tat-βsyn, the Tat-βsyn-degron peptide not only acts as an interference peptide that inhibits α-synuclein oligomerization but also acts as a proteasomal targeting peptide that reduces the level of α-synuclein in neurons. Since α-synuclein is overexpressed under certain pathological conditions of PD[52,53] and these upregulated proteins can interfere with many physiological processes, such as ER-to-Golgi transport[54], synaptic transmission[55], and mitochondria function and morphology[56], robustly knocking down the overexpressed α-synuclein using the Tat-βsyn-degron peptide may have better neuroprotective efficacy in restoring normal cellular functions in the PD brain than simply inhibiting the formation of toxic α-synuclein oligomers using the Tat-βsyn peptide.

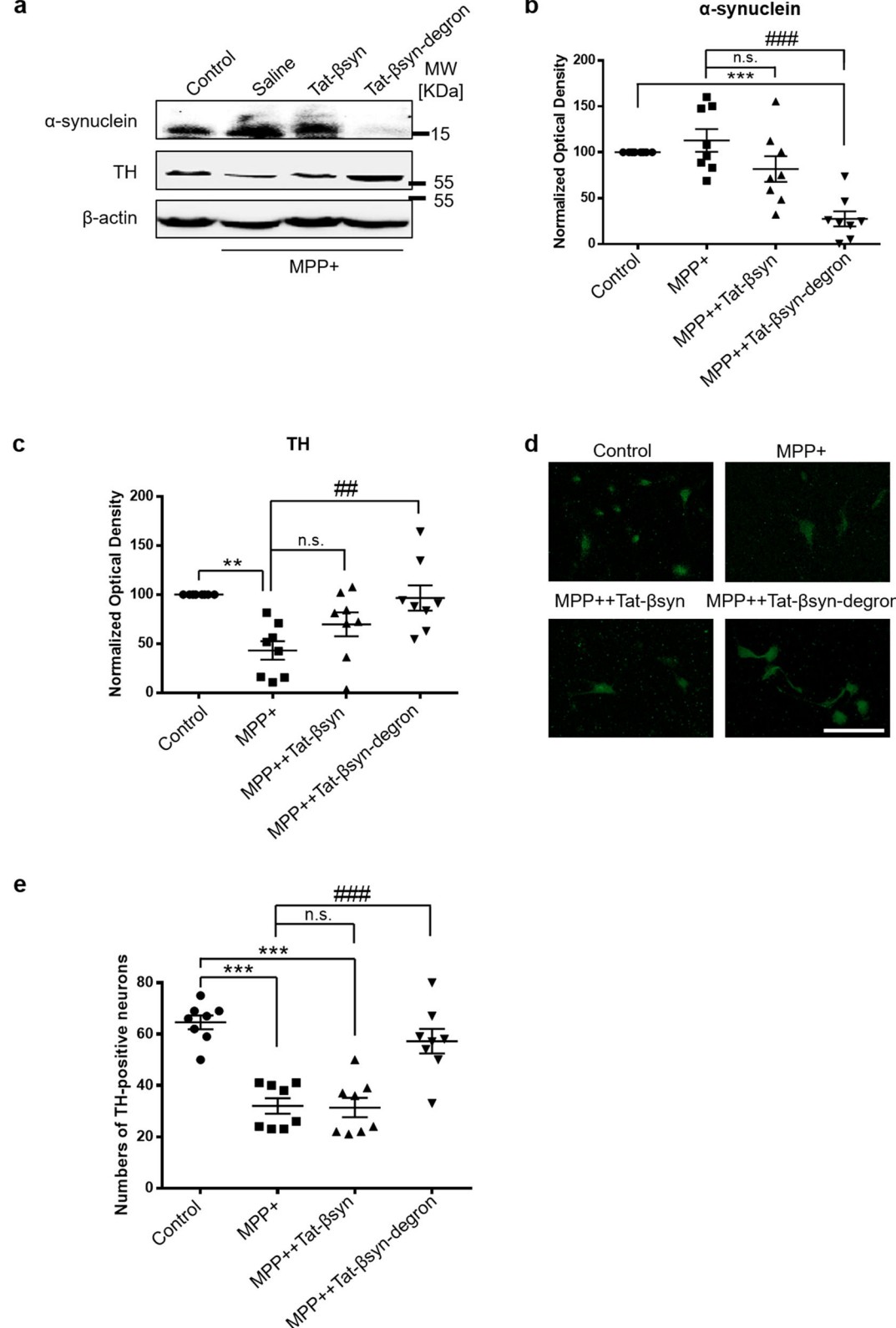

It is also interesting to note that α-synuclein is also expressed in tissues outside the central nervous system and we found that a single i.p. injection of the Tat-βsyn-degron peptide similarly reduced the α-synuclein expression in the kidney and the spleen of wild-type C57BL/6 mice (Supplementary Fig. 5). Thus, whether any of potential therapeutic effects of systemic administration of Tat-βsyn-degron can be attributed to a reduction of α-synuclein in any of the peripheral tissues remains to be further clarified in future studies.

Taken together, the Tat-βsyn-degron peptide shows promising efficacy in knocking down α-synuclein both in vitro and in vivo and in reducing PD pathology in two different animal models of PD. Through further testing and optimization, this peptide may become a new, effective, and clinically practical therapeutic

**Fig. 4 Tat-βsyn-degron peptide protects against parkinsonian toxin-induced neuronal damage in rat ventral midbrain cultures.** Immunoblotting (**a–c**) and immunocytochemical staining (**d**) followed by blinded TH-positive neuron counting (**e**) showing that bath application (25 μM; 48 h) of Tat-βsyn-degron, but not Tat-βsyn, significantly reduced the level of endogenous α-synuclein (**a**, **b**; $N = 8$; $F(3,21) = 18.39$, $P < 0.001$; Bonferroni post hoc test: MPP+ +Tat-βsyn-degron vs. MPP+: ###$P < 0.001$), and prevented MPP+ (20 μM; 48 h)-induced TH-positive neuronal damage as demonstrated by the loss of TH protein (**a**, **c**; $N = 8$; $F(3,21) = 6.87$, $P < 0.01$; Bonferroni post hoc test: MPP++Tat-βsyn-degron vs. MPP+: ##$P < 0.01$) and by the decreased numbers of TH-positive neurons (**e**; $N = 8$; $F(3,21) = 38.00$, $P < 0.001$; Bonferroni post hoc test: MPP++Tat-βsyn-degron vs. MPP+: ###$P < 0.001$). **d** Representative TH-positive neuron staining. Data are presented as mean ± SEM. The statistical difference between groups was determined by two-way ANOVA, followed by Bonferroni post hoc test. **$P \leq 0.01$ and ***$P \leq 0.001$ compared with the control. ##$P \leq 0.01$ and ###$P \leq 0.001$ indicate the statistical difference between the MPP++peptide group and the MPP+ group. n.s. denotes not significant. Scale bar in **d**: 50 μm. Solid circles, squares and triangles represent individual data points in each group.

treatment for PD in human patients. A recent success in a phase 3 clinical trial has already demonstrated that a Tat-fused short peptide is not only safe, but therapeutically effective in protecting neurons against ischemic damage in humans[20]. We hope this α-synuclein knockdown peptide may also have the potential to be quickly translated into the clinic as an effective disease-modifying treatment that directly targets the disease-causing process of PD. Due to the versatility of our peptide-mediated protein knockdown method, we can theoretically target disease-causing cellular proteins by simply changing the protein-binding sequence of the targeting peptide. Since many human diseases, including some of the age-related neurodegenerative diseases such as Alzheimer's disease and Huntington's disease, are pathologically caused by gain of function of a protein due to its mutations and/or increased expression levels, the proposed study can be expected to spur the development of new therapeutics for human diseases beyond PD.

## Methods

**Animals.** Male C57BL/6 mice (20–25 g, purchased from Charles River (Beijing Office, China)) were housed in plastic cages with free access to food and water and maintained in a temperature-controlled room (22 °C) with a 12/12 h light/dark cycle. All experimental protocols were approved by the Chongqing Medical University Animal Care Committee, and the methods were carried out in accordance with the approved guidelines and regulations. All efforts were made to minimize animal suffering and to reduce the number of animals used.

Male M83 hemizygous mice overexpressing the human A53T α-synuclein mutant (12 weeks old, 25–30 g, 004479, The Jackson Laboratory) were housed individually and maintained on a 12/12 h light/dark cycle at 22 °C ambient temperature and with unlimited access to food and water. Housing, breeding, and procedures were performed according to the Canadian Council on Animal Care and were approved by the McGill University Animal Care Committee.

**Chemicals and reagents.** Anti-α-synuclein antibody (BD Transduction Laboratories, 610786), anti-phosphorylated pS129 α-synuclein antibody (ab184674, Abcam), anti-β-actin antibody (Abcam, ab8227), anti-HA antibody (Roche, 118674231001), anti-TH antibodies (BD Transduction Laboratories, 612300, for immunoblotting and immunohistochemistry; Novus, NB300-109, for immunocytochemistry), anti-Iba-1 antibody (019-19741, Wako), anti-GABA$_A$ receptor β2/3 antibody (Millipore, 05-474), anti-HSP90 antibody (BD Transduction Laboratories, 610418), anti-14-3-3 antibody (Millipore, 06-511), MG132 (Sigma, C2211), MPP+ iodide (Sigma, D048), MPTP hydrochloride (Sigma, M0896), normal goat serum (G9023, Sigma-Aldrich), goat anti-rabbit horseradish peroxidase (111-035-144, Jackson ImmunoResearch), goat anti-mouse horseradish peroxidase (115-035-146, Jackson ImmunoResearch). Tat-βsyn-degron peptide (YGRKKRRQRRRRTKSG-VYLVGRRRG), Tat-βsynN-degron peptide (YGRKKRRQRRRRGVLYVGSKTRR RRG), and Tat-βsyn control peptide (YGRKKRRQRRRRTKSGVYLVG) were chemically synthesized by GL Biochem (Shanghai, China). Tat peptide (YGRKKRRQRRR) was synthesized in our lab using the Prelude peptide synthesizer (Protein Technologies Inc.).

**Buffers and media.** PBS (pH 7.4) contained 137 mM NaCl, 2.7 mM KCl, 8.1 mM Na$_2$HPO$_4$, and 1.76 mM KH$_2$PO$_4$. The 1× Tris-buffered saline containing 0.1% Tween-20 (TBST) pH 7.6 contained 20 mM trizma base, 150 mM sodium chloride, and 0.1% Tween-20. Citrate buffer pH 6.0 contained 10 mM tri-sodium citrate. Cell lysis buffer contained 0.5% Triton X-100, 0.5% deoxycholic acid, and 1× protease and phosphatase inhibitor cocktail (Thermo Scientific, 78442) in sterile PBS. The 4× sample buffer contained 50% glycerol, 125 mM pH 6.8 Tris-HCl, 4% sodium dodecyl sulfate (SDS), 0.08% bromophenol blue, and 5% β-mercaptoethanol. Neuron culture media contained 2% B-27 supplement (Invitrogen, 17504-044) and 0.5 mM GlutaMax supplement (Invitrogen, 35050-061) in Neurobasal media (Invitrogen, 21103-049).

**Plasmid construction.** The human α-synuclein plasmid was a generous gift from Dr. Hong Qing from Beijing Institute of Technology (China). The FLAG-βsynN-degron and FLAG-βsyn-degron peptide sequences were translated back to complementary DNA (cDNA) sequences and the corresponding sense and antisense DNA oligonucleotide strands were synthesized by Integrated DNA Technologies (IDT).

FLAG-βsynN-degron (start condon (ATG) and stop codon (TAA) are underlined and bolded):

sense, 5′-AGCTT**ATG**GACTACAAGGACGACGATGACAAGGGGGTGCT GTACGTGGGGGAG**AAG**ACGAGGCGACGACGAGGC**TAA**GC -3′

antisense, 5′-GGCCGCTTAGCCTCGTCGTCGCCTCGTCTTGCTCCCCAC GTACAGCACCCCCTTGTCATCGTCGTCCTTGTAGTCCATA -3′

FLAG-βsyn-degron (start condon (ATG) and stop codon (TAA) are underlined and bolded):

sense, 5′-CCCAAGCTT**ATG**GACTACAAGGACGACGATGACAAGCGTA CTAAATCTGGTGTTTATTTGGTTGGTCGACGACGAGGC**TAA**GCGGCCG CTTTTTTCCTT -3′

antisense, 5′-AAGGAAAAAAGCGGCCGCTTAGCCTCGTCGTCGACCAA CCAAATAAACACCAGATTTAGTACGCTTGTCATCGTCGTCCTTGTAGTC CATAAGCTTGGG -3′

The two strands were then annealed into duplex according to the manufacturer's protocol and inserted into pcDNA3.0 mammalian expression vector following *Hin*dIII and *Not*I double digestion (*Hin*dIII, Thermo Scientific, FD0504; *Not*I, Thermo Scientific, FD0594). The FLAG-βsyn plasmid was constructed by mutating the CGA residues (corresponding to the first arginine residue in the "RRRG" degron peptide sequence) into the stop codon TGA (underlined and bolded in the primer sequence) on the FLAG-βsyn-degron plasmid. FLAG-βsyn point mutation primers (synthesized by IDT): forward, 5′-TATTTGGTT**GA**CGACGAGGCT-3′; reverse, 5′-AGCCTCGTCGTCAACCAAATA-3′. HA-βsyn and HA-γ-synuclein were PCR amplified from rat cDNA library and then inserted into pcDNA3.0 mammalian expression vector following *Bam*HI (Thermo Scientific, FD0054) and *Not*I double digestion.

HA-βsyn primers (synthesized by IDT, start codon (ATG on the forward primer) and stop codon (TTA on the reverse primer, complementary to TAA) are underlined and bolded):

forward, 5′-CGGGATCC**ATG**TACCCATACGATGTTCCAGATTACGCTAT GGACGTGTTCATGAAGG**CC**TGTCCATG-3′

reverse, 5′-AAGGAAAAAAGCGGCCG**TTA**CGCCTCTGGCTCGTATTCC TGATATTCCTC-3′

HA-γ-synuclein primers (synthesized by IDT, start codon (ATG on the forward primer) and stop codon (CTA on the reverse primer, complementary to TAG) are underlined and bolded):

forward, 5′-CGGGATCC**ATG**TACCCATACGATGTTCCAGATTACGCTAT GGACGTCTTCAAGAAAGG**C**TTCTCCATT-3′

reverse, 5′-AAGGAAAAAAGCGGCCGC**CTA**GTCTCCTCCACTCTTGGCC TCTTCGCCCTC-3′

All plasmid sequences were verified by DNA sequencing.

**HEK 293 cell culture and plasmid transfection.** HEK 293 cells that are commonly used for plasmid transfection and gene expression were purchased from ATCC (ATCC® CRL-1573™) and no mycoplasma contamination were observed during the experiment. HEK 293 cells were cultured in Dulbecco's modified Eagle's medium (Sigma, D6429) supplemented with 10% fetal bovine serum (Invitrogen, 12483020). When HEK 293 cells achieved 90% confluence, plasmids were transfected into the cells using Lipofectamine 2000 (Invitrogen, 11668019) according to the manufacturer's instruction. Total plasmid transfection amount in every group was made equal by supplementing pcDNA3.0 empty vector. HEK 293 cells were then maintained in the 37 °C incubator with 95% O$_2$ and 5% CO$_2$ for 48 h before being used in experiments.

**Biacore peptide–protein binding assay.** Biacore experiments were performed using a Biacore 3000 instrument (GE Healthcare Biosciences, Upsala, Sweden) and HBS running buffer, pH 7.4, containing 10 mM HEPES, 150 mM NaCl, 3 mM

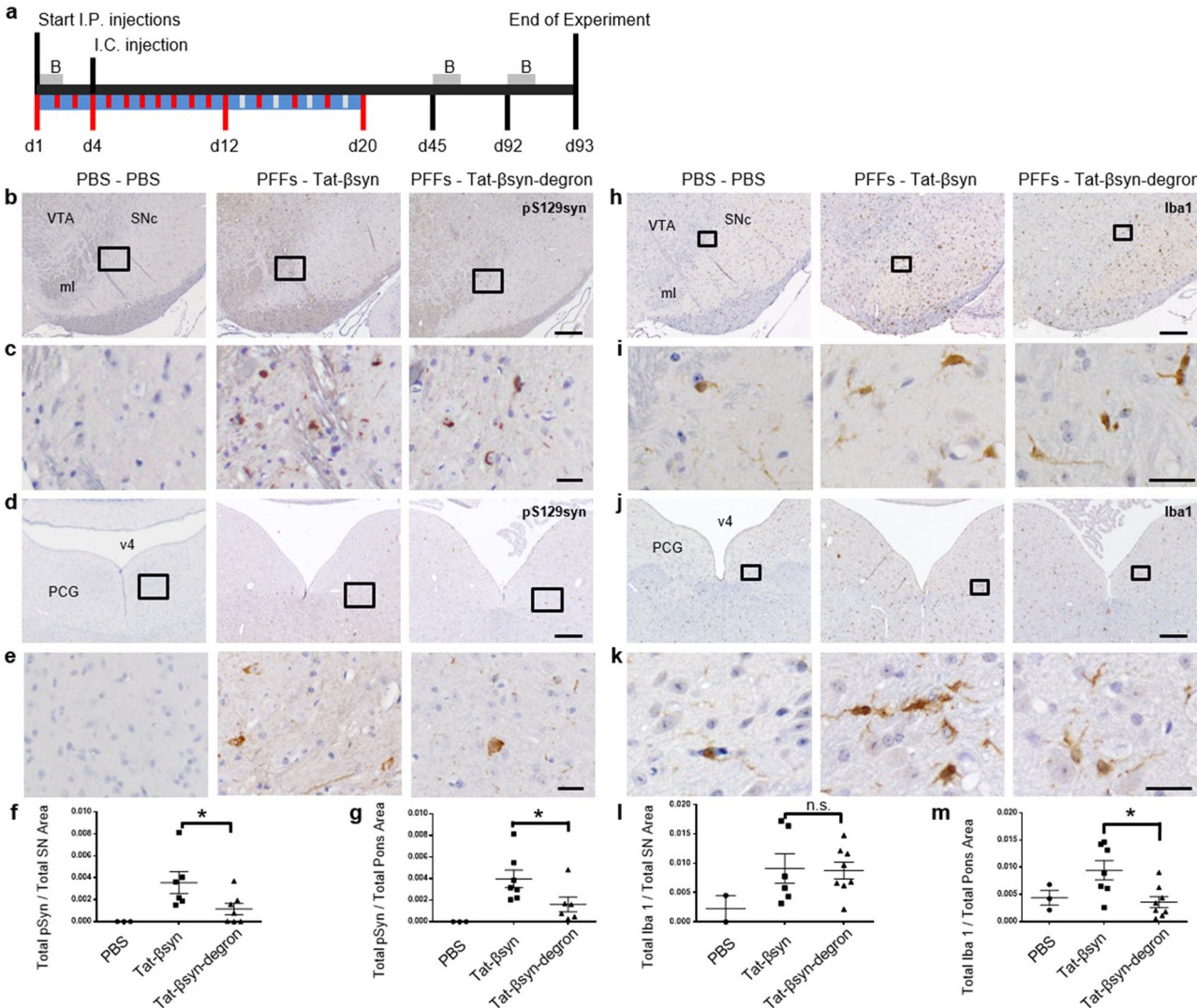

**Fig. 5 Tat-βsyn-degron protects against α-synuclein propagation in a mouse model of synucleinopathy. a** The experimental timeline. M83 mice were injected i.c with PBS or PFFs and i.p. with PBS, Tat-βsyn, or Tat-βsyn-degron. B: behavioral tests. Vertical red lines: the days that the mice received i.p. injections. Vertical white lines: the days off i.p. injections. **b, d** Phosphorylated (pS129) α-synuclein immunostaining of coronal sections at the level of substantia nigra (**b**) and pons (**d**) from M83 mice. **c, e** Magnified images expanded from boxes in **b** or **d** showing pS129 α-synuclein staining. **f, g** Histogram showing the total pS129 α-synuclein (Total pSyn) staining per area at the level of substantia nigra (**f**, PBS: $N = 3$; Tat-βsyn: $N = 6$; Tat-βsyn-degron: $N = 7$; $t = 2.235$, d.f. $= 11$, $P < 0.05$) and pons (**g**, PBS: $N = 3$; Tat-βsyn: $N = 7$; Tat-βsyn-degron: $N = 6$; $t = 2.180$, d.f. $= 11$, $P = 0.05$) quantified from panels (**b, d**). **h, j** Microglial Iba-1 immunostaining of coronal sections at the level of substantia nigra (**h**) and pons (**j**) from M83 mice. **i, k** Magnified images expanded from boxes in **h** and **j** showing Iba-1 staining. **l, m** Histogram showing the total Iba-1 staining per area at the level of substantia nigra (**l**, PBS: $N = 2$; Tat-βsyn: $N = 6$; Tat-βsyn-degron: $N = 8$; $F(2,13) = 1.61$, $P = 0.238$) and pons (**m**, PBS: $N = 3$; Tat-βsyn: $N = 7$; Tat-βsyn-degron: $N = 8$; $F(2,15) = 5.14$, $P < 0.05$, Bonferroni post hoc test: Tat-βsyn-degron vs. Tat-βsyn: $P < 0.05$) quantified from panels (**h, l**). Data are presented as mean ± SEM. Statistical significance in **f** and **g** was determined by unpaired $t$ test (because the PBS group did not have any pS129 α-synuclein staining and all the values were 0). Statistical significance in **l** and **m** was determined by one-way ANOVA, followed by Bonferroni post hoc test. *$P \leq 0.05$ denotes the difference between the Tat-βsyn group and the Tat-βsyn-degron group. n.s. denotes not significant. ml medial lemniscus, PCG pontine central gray, SNc subtantia nigra pars compacta, v4 4th ventricle, VTA ventral tegmental area. Scale bar: 200 μm in panels (**b, d, h, j**); 25 μm in panels (**c, e, i, k**). Solid circles, squares and triangles represent individual data points in each group.

EDTA, and 0.005% surfactant P20. A research-grade CM5 sensor chip was activated with a mixture containing equal molar amounts of EDC (N-ethyl-N'-(dimethylaminopropyl) carbodiide) and NHS (N-hydroxysuccinimide). Purified recombinant human α-synuclein (N-terminal histidine tagged, Sigma, S7820-500UG), diluted in sodium acetate pH 3.1, was then injected and covalently coupled to a flow cell of the sensor chip surface by amide bonding. Residual unreacted sites were blocked with ethanolamine. A reference surface, to account for non-specific binding, was similarly generated by activating and blocking an adjacent flow cell. Approximately 1500 resonance units (RUs), equivalent to a surface concentration of 1500 pg/mm² of α-synuclein, was immobilized.

To ascertain the binding interaction between α-synuclein knockdown peptides and α-synuclein, synthetic Tat-βsyn-degron peptide or control Tat peptide was serially diluted and sequentially injected over the active surface containing immobilized α-synuclein and the reference surface for 3 min at a flow rate of 30 μl/min. The peptides were then allowed to dissociate for 3 min during which time HBS running buffer was injected. The resultant sensorgrams were double-referenced by subtracting out the binding on the reference surface and the response from the HBS blank buffer control. Binding response report points were collected 20 s into the dissociation phase of the interaction at time 200 s, which represents a stable binding response and excludes bulk refractive index changes and nonspecific binding.

**Primary neuron culture**. Primary rat neuron cultures were prepared from embryos of pregnant Sprague–Dawley rats (E18). The experimental protocol was approved by the University of British Columbia Animal Care Committee. Briefly, the cortical

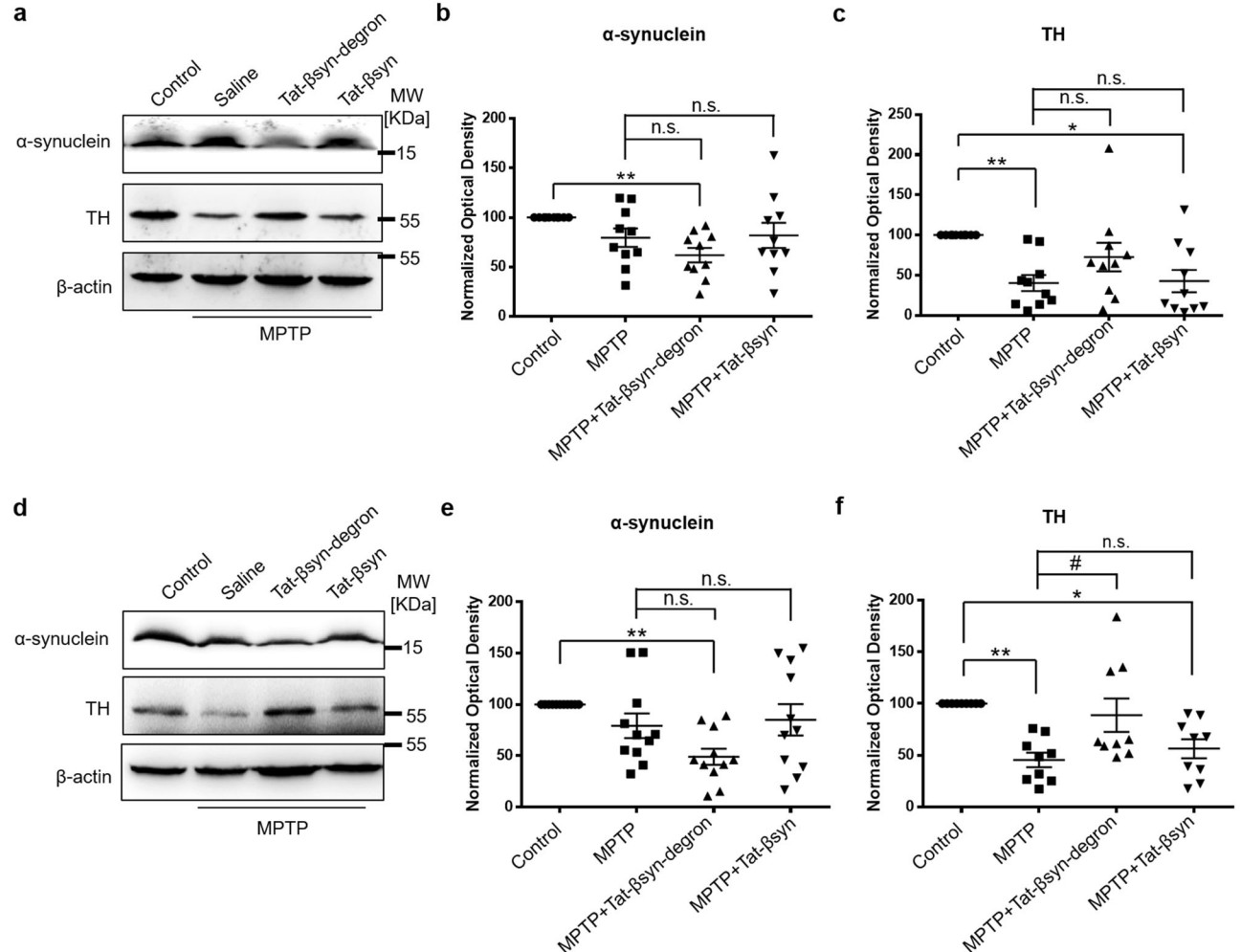

**Fig. 6 Tat-βsyn-degron peptide-mediated knockdown of α-synuclein protects against parkinsonian toxin MPTP-induced TH protein decrease. a–f** Mice received i.p. injections of MPTP (30 mg/kg) or same volumes of saline once a day for 5 days, along with Tat-βsyn-degron or its control Tat-βsyn (6 μmol/ kg; i.p.) twice a day for 12 days. Brain tissues were collected for immunoblotting for α-synuclein and TH immediately after behavioral assessments on day 12. Tat-βsyn-degron, but not Tat-βsyn, significantly reduced α-synuclein in the substantia nigra-containing ventral midbrain (**a**, **b**; $N = 10$; $F(3,27) = 5.28$, $P < 0.01$; Bonferroni post hoc test: MPTP+Tat-βsyn-degron vs. MPTP: $P = 0.461$) and the striatum (**d**, **e**; $N = 11$; $F(3,30) = 7.24$, $P < 0.01$; Bonferroni post hoc test: MPTP+Tat-βsyn-degron vs. MPTP: $P = 0.070$), and protected against the MPTP-induced decrease in the level of TH protein in both ventral midbrain (**a**, **c**; $N = 10$; $F(3,27) = 5.97$, $P < 0.01$; Bonferroni post hoc test: MPTP+Tat-βsyn-degron vs. MPTP: $P = 0.349$) and striatum (**d**, **f**; $N = 9$; $F(3,24) = 8.33$, $P < 0.01$; Bonferroni post hoc test: MPTP+Tat-βsyn-degron vs. MPTP: #$P < 0.05$). Data are presented as mean ± SEM. The statistical difference between groups was determined by two-way ANOVA, followed by Bonferroni post hoc test. *$P \leq 0.05$ and **$P \leq 0.01$ compared with the control. #$P \leq 0.05$ indicates the statistical difference between the MPTP+peptide group and the MPTP group. n.s. denotes not significant. Solid circles, squares and triangles represent individual data points in each group.

tissue or ventral midbrain tissue was isolated into ice-cold Hanks' balanced salt solution (Invitrogen, 14170-112) and then digested with 0.25% trypsin-EDTA at 37 °C for 30 min. After washing with warm Dulbecco's modified Eagle's medium (supplemented with 10% fetal bovine serum) three times, neurons were suspended in neuron culture media and dissociated by trituration using varying sizes of pipettes. Neurons were then centrifuged, and the pellet was resuspended in culture media, washed twice with culture media, and plated on the poly-D-lysine-coated plates. Neuron culture was maintained in the 37 °C incubator with 95% $O_2$ and 5% $CO_2$. The morning after culturing, 2/3 of the neuron culture media were replaced with fresh neuron culture media. Media were then replaced every 3–4 days. Primary rat cortical neuron culture was used 14 days in vitro (DIV) and primary rat ventral midbrain neuron culture was used 3 DIV.

**Peptide and MPP+ treatment.** Tat-βsyn-degron and Tat-βsyn peptides were first dissolved in sterile water as a 25 mM stock solution and then diluted directly in the neurobasal culture media to make the desired working concentration. Twenty micromoles of MPP+ iodide stock solution was made freshly each time and diluted in the neurobasal culture media directly to make the desired working concentration. For 48 h treatment, neuron culture media containing MPP+ and peptides were replaced every 24 h.

**Cytotoxicity.** DIV 14 primary cortical neurons were treated with different doses of the Tat-βsyn-degron and Tat-βsyn peptides and culture media were collected 24 h after peptide treatment to measure cytotoxicity using an LDH (lactate dehydrogenase) assay kit (Roche, 11644793001). The culture medium from cells treated with 2% Triton X-100 for 30 min at 37 °C was used as the positive assay control and the culture medium from untreated cells were used as the negative assay control.

**TH-positive neuron counting in vitro.** Neurons were rinsed 4× with ice-cold PBS, 2 min each time, and fixed with 4% PFA for 1 h at 37 °C. Neurons were then washed 3× in PBS for 5 min with gentle agitation, and subsequently incubated in 0.25% Triton X-100/PBS for 5 min at room temperature with gentle shaking. Next, neurons were washed 1× in PBS for 5 min, and then incubated for 30 min at 37 °C in 10% bovine serum albumin (BSA)/PBS without agitation to block nonspecific staining. To label TH, neurons were incubated in primary TH antibody (1:100 dilution in 3% BSA/PBS) at 4 °C for 5 days without agitation. Neurons were then washed 6× in PBS for 2 min, and incubated in Alexa Fluor 488 (Life Technologies, A-11034; 1:500 dilution in 3% BSA/PBS) for 45 min at 37 °C without agitation. Next, neurons were washed 6× in PBS for 2 min, mounted on glass slides with Fluoromount-G slide mounting media (SouthernBiotech, 0100-01), and stored at

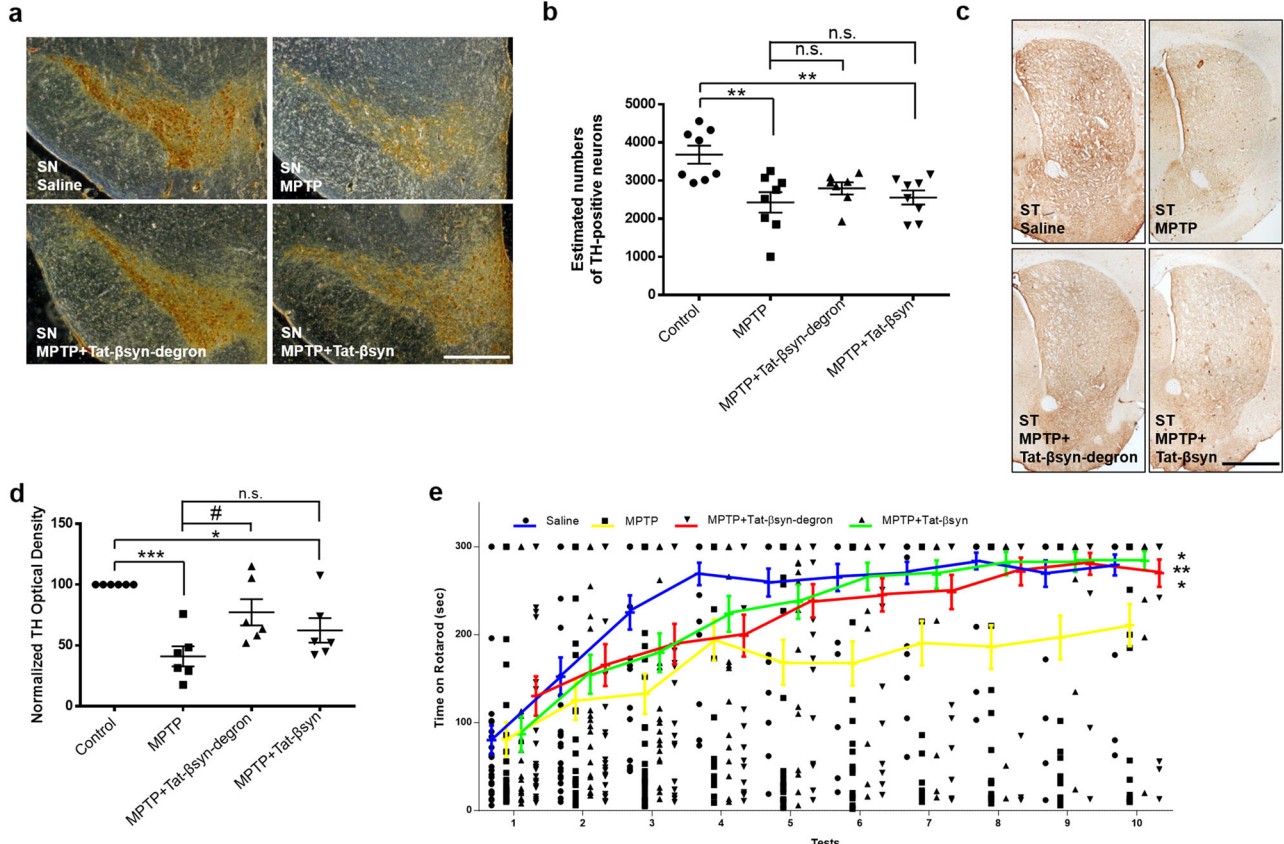

**Fig. 7 Tat-βsyn-degron peptide-mediated knockdown of α-synuclein protects against parkinsonian toxin MPTP-induced dopaminergic neuronal damage and behavioral deficits in mice. a–e** Mice received i.p. injections of MPTP (30 mg/kg) or same volumes of saline once a day for 5 days, along with Tat-βsyn-degron or its control Tat-βsyn (6 μmol/kg; i.p.) twice a day for 12 days. Brain tissues were collected for immunohistochemical staining of TH (**a**, **c**) immediately after behavioral assessments (**e**) on day 12. **a–d** Tat-βsyn-degron, but not its control Tat-βsyn, protected against MPTP-induced decrease in the number of dopaminergic neurons in the substantia nigra (SN) pars compacta as estimated by blinded neuron counting from bregma −2.92 to −3.64 mm (**b**; $N = 7$ for the Tat-βsyn-degron group and $N = 8$ for the other three groups; $F(3,20) = 7.02$, $P < 0.01$; Bonferroni post hoc test: MPTP+Tat-βsyn-degron vs. MPTP: $P = 1.00$) and the density of dopaminergic neuronal terminals in the striatum (ST; **d**; $N = 6$ for all groups; $F(3,15) = 12.29$, $P < 0.001$; Bonferroni post hoc test: MPTP+Tat-βsyn-degron vs. MPTP: $^{\#}P < 0.05$). **e** Rotarod motor behavioral tests revealed that mice treated with chronic MPTP (MPTP; $N = 28$) showed significantly shorter latency in falling off the rotarod compared with mice receiving saline control (Saline; $N = 28$), and that the MPTP-induced motor deficits were significantly reduced by Tat-βsyn-degron (MPTP+Tat-βsyn-degron; $N = 28$) and by Tat-βsyn (MPTP+Tat-βsyn; $N = 28$) ($F(3,108) = 4.94$; $P < 0.01$; Bonferroni post hoc test: compared with the MPTP group: saline, $^{**}P < 0.01$; MPTP+Tat-βsyn: $^*P < 0.05$; MPTP+Tat-βsyn-degron: $^*P < 0.05$). All data are presented as mean ± SEM. The statistical difference between groups was determined by two-way ANOVA, followed by Bonferroni post hoc test. $^*P ≤ 0.05$, $^{**}P ≤ 0.01$, and $^{***}P ≤ 0.001$ compared with the control in panels (**b**, **d**) and with the MPTP group in panel (**e**). $^{\#}P ≤ 0.05$ indicates the statistical difference between the MPTP+peptide group and the MPTP group. n.s. denotes not significant. Scale bar: 0.5 mm in panel (**a**) and 1 mm in panel (**c**). Solid circles, squares and triangles represent individual data points in each group.

room temperature overnight to dry. Neurons were then imaged with the Zeiss Axio Observer D1 microscope at ×20 and ten fields of view per coverslip were randomly selected and counted. Imaging and counting for TH-positive cells were performed by an experimenter blinded to the treatment conditions.

**Preformed fibrils.** PFFs were generated in-house based on the Volpicelli-Daley et al. protocol[57]. Synuclein monomers were shaken at 37 °C at 1000 r.p.m. for 7 days. PFFs were sonicated with 60 pulses at 10% power (total of 30 s, 0.5 s on, 0.5 s off; FB120, Fisher Scientific), stored at −80 °C, and kept at room temperature during the intracerebral injections. PFFs were characterized using a negative staining protocol and analyzed using an electron microscope. PFFs were added to 200 mesh cupper carbon grid (3520C-FA, SPI Supplies) and stained with 2% acetate uranyl (22400-2, EMS). PFFs were visualized using a transmission electron microscope (FEI Tecnai 12 Bio Twin 120 kV TEM) coupled to a AMT XR80C CCD camera, and analyzed with ImageJ 1.5 and Matlab 2017b software (Supplementary Fig. 2a, b).

**Intracerebral injection in M83 mice.** M83 mice were anesthetized with 2% iso-flurane and underwent stereotaxic injection with one of the following inoculants: human α-synuclein PFFs (total protein concentration, 12.5 μg per brain) or PBS, at a rate of 0.25 μl/min (for a total volume of 2.5 μl). A dose of 20 mg/kg carprofen and 250 mg/ml bupivacaine were administered subcutaneously to the mouse prior

to the craniotomy. Five microliters of Hamilton syringe with a 33-gauge needle was placed unilaterally in the right dorsal striatum (+0.2 mm relative to bregma, +2.0 mm lateral from midline, and 2.6 mm ventral from skull dorsal surface)[29] using the mouse brain in Stereotaxic Coordinates Atlas[58]. Mice received intraperitoneal injections of PBS, the Tat-βsyn-degron peptide (40 mg/kg), or the Tat-βsyn peptide (40 mg/kg), daily for 12 days, including 3 days before and 9 days after the i.c. injection, and subsequently every other day for 8 days (20 days in total, Fig. 5a).

**Enzyme-linked immunosorbent assay.** Brains were lysed (1 g tissue per 3 mL solution) in a buffer solution (50 mM Tris pH 8, 150 mM NaCl, 5 mM EDTA, 1% NP-40, 0.5% sodium deoxycholate, 0.1% SDS, protease and inhibitor cocktail [aprotinin, leupeptin, and benzamidine], and phosphatase inhibitor cocktail) on ice. Lysates were homogenized at $1600 \times g$ several times, sonicated, and finally centrifuged at $100,000 \times g$ for 30 min. The supernatant was collected and processed with the human α-synuclein ELISA kit (KHB0061, Thermo Fisher Scientific) and analyzed with a Tecan microplate reader (Tecan Infinite M200 Pro, life Science). The level of α-synuclein was expressed as ng/ml total sample.

**Mouse MPTP in vivo model of PD.** C57BL/6 mice (20–25 g) received intraper-itoneal injection of 30 mg/kg parkinsonian toxin MPTP hydrochloride once a day for 5 days to induce dopaminergic neuron death in the substantia nigra, while the control mice received equal volumes of saline injection. Six micromoles per kg of

Tat-βsyn-degron peptide or control Tat-βsyn peptide was i.p. injected into the MPTP-treated mice every 12 h from the first day of MPTP injection until 7 days after the last injection of MPTP. All groups of mice then underwent a rotarod test before they were sacrificed.

**Behavioral tests**. The rotarod test was performed as previously described, with modifications[34]. Briefly, 6 days after the last MPTP injection, all mice received four rounds of training on the rotarod (Stoelting Co.). In the first two rounds of training, the rotarod was maintained at a constant speed of 20 r.p.m. for 3 min. In the second two rounds of training, the rotarod reversed rotation direction every three turns at the constant speed of 20 r.p.m. for 3 min. Twenty-four hours after the last round of training, all groups of mice received formal rotarod testing in which the rotarod reversed rotation direction every three turns at the constant speed of 20 r.p.m. Mice were tested ten times at 20 min intervals, and the time that they remained on the rotarod during each test was recorded. Maximum test time (cut-off limit) was 300 s. The motor performance of the mouse was expressed as the latency to fall off the rotarod.

In order to study motor function in M83 mice, rotarod and wire-hang tests were performed at days 1, 45, and 92 of the experiment, in mice that had been previously trained for the tests. For the rotarod test (Bioseb), the speed of the rotating rod gradually increased by 40 r.p.m. every 30 s, for a maximal time of 5 min. For the wire-hang test, a wire (diameter between 2 and 3 mm) was placed horizontally in a plastic box (length 52 cm, width 38 cm, height 34 cm). The mouse was placed on the wire hanging upside down. The latency of fall from the wire was recorded up to a maximal time of 3 min. The behavioral tests were performed in triplicate and the best performance was used for statistical analysis.

**Immunoblotting**. Brain tissues or cultured cells were lysed on ice in the lysis buffer and then the solution was centrifuged at 14,000 r.p.m. for 10 min at 4 °C. Next, the supernatant was collected and protein concentrations were determined using a BCA protein assay kit (Thermo Scientific, 23227). Equal amount of protein samples was mixed with 4× sample buffer, boiled at 100 °C for 5 min, and separated on 10% SDS-polyacrylamide gel electrophoresis. Proteins were then transferred to Immobilon-PTM polyvinylidene fluoride membranes (Bio-Rad, 162-0177). The membranes were blocked with 5% non-fat milk in TBST for 1 h at room temperature, and then incubated overnight at 4 °C with primary antibody. After washing 3× in TBST for 5 min, membranes were incubated with horseradish peroxidase-conjugated secondary antibody for 1 h at room temperature. After another three washes with TBST, protein was visualized in the Bio-Rad Imager using ECL western blotting substrate (Pierce, 32016). The band density of each protein was quantified by the Bio-Rad Quantity One software and the relative optical density was analyzed relative to the loading control β-actin on the same membrane.

**Immunohistochemical staining**. C57BL/6 mice were anesthetized with 1.5 g/kg urethane (Sigma, U2500) and then perfused with 0.9% saline and 4% paraformaldehyde. Brains were removed and fixed in 4% paraformaldehyde for 24 h before being transferred to 30% sucrose/PBS solution for cryoprotection. The brains were then sectioned into 30 μm slices using a Leica cryostat. After blocking and permeabilization using a PBS solution containing 1% BSA and 0.2% Triton X-100 for 30 min at room temperature, the brain slices were incubated with anti-TH antibody (1:800 dilution) for 3 days at 4 °C. Finally, the TH-positive neurons in the substantia nigra and TH-positive neuronal terminals in the striatum were stained using the anti-mouse Ig HRP detection kit (BD Transduction Laboratories, 551011) according to the manufacturer's instruction, and visualized under Zeiss Axio Observer D1 microscope.

Brains from M83 mice that received i.c. and i.p. injections were processed for immunohistochemistry. Mice were anesthetized with 2% isoflurane and were perfused with 10% formalin i.c. The brains were removed and postfixed with 10% formalin for 24 h at 4 °C. Coronal sections were cut with a paraffin microtome at 5 μm thickness. Tissue sections were incubated in citrate buffer pH 6.0 for 10 min, then rinsed with TBST, and finally incubated in 3% oxidase peroxidase for 15 min. Subsequently, the sections were blocked with 10% normal goat serum in TBST for 30 min at room temperature and incubated with the following primary antibodies: anti-phosphorylated pS129 α-synuclein (1:500 dilution) or anti-Iba-1 (1:250 dilution) in 5% normal goat serum in TBST, overnight at 4 °C. Afterwards, the sections were incubated in secondary antibodies, goat anti-mouse-HRP (1:500 dilution) or goat anti-rabbit-HRP (1:500 dilution), for 30 min at room temperature. The peroxidase reaction product was visualized as a brown precipitate by incubation of the tissue with the DAB substrate kit (8059, Cell Signal Technology). Immunohistochemistry sections were examined by bright field microscope (Olympus DP-21SAL coupled to a digital camera DP21/DP26). Coronal sections were analyzed using the Fiji-ImageJ 1.5 software to detect the total area labeled with the peroxidase immunoreaction product (Supplementary Fig. 2c). The results were analyzed using the GraphPad Prism software.

**Densitometric analysis of striatal TH staining**. The optical density of the TH-positive neuronal terminal staining in the mouse dorsolateral striatum where dopaminergic inputs from substantia nigra pars compacta were received[33] was quantified using the NIH ImageJ software. The optical density from the overlying corpus callosum was used as a background[59] and subtracted from every measurement in the striatum. The optical density in the experimental group was normalized to the value from the control group.

**Dopaminergic neuron counting in the mouse substantia nigra pars compacta**. Mouse substantia nigra was sliced into a series of 30 μm sections (from rostral to caudal), and one in every six sections was stained with anti-TH antibody. The substantia nigra pars compacta was outlined using TH-positive neurons between identifiable landmarks[60] from bregma −2.92 to −3.64 mm. Identifiable TH-positive neurons in three stained sections within this distance from each animal were manually counted by an experimenter blinded to the treatment conditions under the Zeiss Axio Observer D1 microscope at ×40, and the total number of TH-positive neurons in this region was estimated using the following equation: $N =$ counted number × 6.

**Statistical and reproducibility**. Cell cultures and animals were randomly chosen for experiments and the data were analyzed by the SPSS software and expressed as mean ± SEM. Bar graphs were shown in dot-plot format in order to show data distribution. Measurements were taken from distinct samples for statistical analysis. Data were analyzed by one-way analysis of variance (ANOVA), two-way ANOVA, or unpaired $t$ test, depending on the experiment condition. ANOVA analyses were followed by Bonferroni or Tukey's HSD (honestly significant difference) post hoc test, depending on the number of groups under investigation. The sample sizes and number of replicates were indicated in the figure legends. $*P \leq 0.05$, $**P \leq 0.01$, and $***P \leq 0.001$ were considered as significant differences.

**Reporting summary**. Further information on research design is available in the Nature Research Reporting Summary linked to this article.

## Data availability

All source data underlying the graphs presented in the main and Supplementary figures are available as Supplementary data in Excel format. The full, uncropped blot/gel images are shown in Supplementary Fig. 6. All other data supporting the findings of this study are available within the paper and the Supplementary information.

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

## Acknowledgements

This work was supported by grants from the National Natural Science Foundation of China (Nos. 91749116, 81622015, and 82071395) and the Chongqing Science and Technology Commission (No. cstc2020jcyj-zdxmX0004) to Z.D., the Canadian Institutes of Health Research (FDN-154286), Brain Canada, and The Weston Brain Institute research grant to Y.T.W. and a CIHR Foundation grant (FDN—154301) to E.A.F. E.A.F. is the holder of a Canada Research Chair (Tier 1) in Parkinson Disease. Y.T.W. is the holder of Heart and Stroke Foundation of British Columbia and Yukon Chair in Stroke Research. We thank Dr. Vincent Georges Charles Soubannier for helping with the immunohistochemistry analysis. We thank Yuping Li and Kimberley Girling for cell culture support. We thank Dr. Loren Oschipok and Ms. Rebecca Wiens for their excellent editorial assistance.

## Author contributions

J.W.J. and X.F. conceived the idea, designed experiments, conducted all the in vitro experiments and some of the in vivo experiments, and performed most of the data analysis. L.Z., C.D., X.W., H.L., and W.H. conducted the rotarod behavioral test and did the immunostaining of the MPTP mouse brain tissues. E.G. and N.C. conducted the Biacore binding assay. A.H. and B.R.L. assisted in blinded neuron counting in slices. J.L. assisted in plasmid constructions and experiment design. L.L. assisted in peptide synthesis and experiment design. E.d.C.-P. and X.-X.L. conducted the in vivo experiments in M83 mice. E.d.C.-P. performed the data analysis in M83 mice. E.A.F. and T.M.D. designed and supervised M83 mice in vivo experiments. Z.D. supervised the rotarod behavioral test and the data analysis of the MPTP mouse brain tissues. Y.T.W. conceived the ideas, designed the experiments, supervised the whole project, and wrote the manuscript with J.W.J. and E.d.C.-P.

## Competing interests

The authors declare no competing interests.
