## [Peer Review File · Communications Biology]

This manuscript has been previously reviewed at another Nature Research journal. This document only contains reviewer comments and rebuttal letters for versions considered at Communications Biology.

Reviewers' comments:

Reviewer #1 (Remarks to the Author):

In this manuscript, Jin, Fan and Cid-Pellitero with colleagues describe the modification of a previously developed (Fan et al., Nat Neurosci. 2014) blood-brain barrier and plasma membrane-permeable α -synuclein knockdown peptide, Tat- β syn-degron. They describe how the peptide is able to reduce α -syn levels via proteasomal degradation in cell and animal models. Specifically, the peptide is able to reduce α -syn levels and microglial activation in a pre-formed fibril model of α -syn propagation in human A53T overexpressing mice. In a MPTP -based PD model, the peptides reduce motor dysfunction. Overall, this is an interesting paper that contributes to furthering the knowledge of targeting α -syn to the field. However, I have some comments regarding different aspects of the manuscript that need to be addressed.

1. How do you know the reduction peaks at 24h (Fig 3B) when no timepoint after 24h is shown here? In vivo data from M83 line confirms this statement, but no such conclusions can be drawn from 3B alone.

2. Fig 5, is it possible to add a visual to clarify experimental timeline? Text from the result section and the methods do not seem to match.

Line 173-176: Starting three days prior to the PFF injection, mice were treated daily for 9 days with either Tat- β syn or Tat- β syn-degron peptide (40 mg/kg; i.p.). Subsequently, the mice were treated with the same dose every other day for an additional 8 days."

Line 500-503: Mice received intraperitoneal (i.p) injections of PBS, Tat- β syn-degron peptide (40 mg/kg) or Tat- β syn peptide (40 mg/kg), daily for 12 days, starting 3 days before the intracerebral (i.c.) injection, and subsequently every other day for 8 days (20 days in total)."

3. PFF model – the brain regions evaluated, how are they connected to site of injection (i.e., how many synapses apart) and each other?

4. Why different treatment regime for the MPTP model vs the PFF model? Injected with the drug at different concentration (6 μ mol/kg MPTP and 40mg/kg PFF) and twice per day for the MPTP model.

5. Fig 6 – As I interpret it, all significance shown is referring to significant differences between the control group and, like in 6B, the MPTP/Tat- β syn-degron group. I think this could be clarified in the main text, only the last sentence in the figure legend explains that everything is rereferring back to the control. I.e., in none of the graphs are there any comparisons being made between the MPTP group and the MPTP/peptide group.

Confusing to reader when Fig 7E is referred to in the figure legend for Fig 6.

6. How is lowering α -syn levels protecting dopaminergic neurons in the MPTP model; it's not known to robustly induce α -syn pathology so what is the mechanism of action here? The authors mention α -syn oligomerization in the discussion, so it would be very interesting to look at oligomerized synuclein to further 1) justify using the MPTP model for evaluating a α -syn targeting drug candidate and 2) characterize the peptides effect (both Tat- β syn or Tat- β syn-degron)?

7. In the methods it is stated that the TH neurons were manually counted in three nigral sections within a set distance. This would preferably be done using stereology to recapture the most accurate,

unbiased cell count.

In addition, the methods describe that the number of cells counted in these three nigral sections were multiplied by 6. I assume this is because only 1/6 of the brain was stained. However, the distribution of cells is not the same throughout the entire nigra, thus multiplying by 6 will by no means give an accurate estimate of the total number of cells.

In addition, why was only three nigral sections counted? In a 6-series of a mouse brain sectioned in 30um intervals, around 6-7 sections should contain the nigra, counting all of those would give a more accurate estimate.

8. For the statistical analysis, only for the rotarod data was analyzed using a two-way ANOVA (as is appropriate). The remainder of the data was analyzed using a student's t-test (for aSyn data in fig 4, also appropriate since the PBS group can be excluded). However, one-way ANOVA and following post hoc tests were performed for the remainder of the data; only comparing the control group with the additional groups. In fig 6F and 7D, # is indicating a trend between the MPTP and the MPTP/ Tat- β syn-degron (not mentioned in the text nor the figure legend). But this cannot be accurately evaluated using a one-way ANOVA since there are two main effects evaluated; 1) if MPTP has an effect and 2) if the treatment with the peptide has an effect, which requires a two-way ANOVA to address.

Reviewer #2 (Remarks to the Author):

In this manuscript authors have evaluated the efficacy of a α -syn-binding synthetic peptide in mitigating the PD-like neuropathology in mice. Study has utilized both in vitro and in vivo approaches. This manuscript can be published with minor revisions.

1. Title, Summary and Discussion. Title of the manuscript is misleading. It should reflect the content of the manuscript. Authors show that their peptide is most effective in MPTP-induced mice. They did not test A53T mice. Their experiment with α -syn spreading model was not successful. The abstract and discussion should also modify accordingly.
2. Fig. 6. In this fig, authors show that β -syn-degron reduces α -syn and mitigates neuronal loss. This observation is consistent with previous reports that suggest that reducing α -syn is beneficial in PD-mouse models. However, this Fig. shows that MPTP does not alter the α -syn level in the brain, this appears to contradict previous reports.

Reviewer #3 (Remarks to the Author):

This study focused on the development of an α -synuclein knockdown peptide, Tat- β syn-degron. By utilizing both in vitro and in vivo Parkinson's disease (PD) models, they observed a decrease of α -synuclein in Tat- β syn-degron while decreasing neuronal damage and motor impairment. They authors concluded that this peptide could lead to a therapeutic treatment in PD. This is a nice paper showing the potential for this technique to reduce alpha-synuclein and possibly protect against progression in PD. Also, I think this technique has a lot of potential as a tool in the lab for reducing protein expression or targeting specific domains within proteins.

In the cortical neurons and the mouse brain, are levels of beta-synuclein reduced? This may be

difficult to show by Western blot if the authors cannot find a good beta-synuclein specific antibody. Could the authors use qPCR to show if beta-synuclein is reduced in the neurons with exposure to the TAT-betasyn-degron?

Examining gamma-synuclein is not necessary because it is not highly expressed in neurons.

I am unclear why the authors used beta-synuclein as targeting sequence besides that a previous study showed this sequence binds to alpha-synuclein. It makes sense as a proof-of-principle that this technique can work. However, I would like to see a targeting sequence specific for alpha-synuclein such as starting around amino acid 72- for future studies at least.

It looks like in primary neurons, there is a reduction in beta-actin with higher concentrations or longer time points of exposure with the TAT-betasyn-degron. It would be nice to see a toxicity assay with the TAT-betasyn-degron compared to the control peptide and no treatment.

For figure 3 and figure 5- did the authors use a human specific synuclein antibody to confirm reduced levels of alpha-synuclein with TAT-betasyn-degron knockdown? Or did they use an antibody that recognizes both mouse and human synuclein? Did the authors perform western blots from forebrain homogenates from these mice? Did the TAT-betasyn-degron reduce alpha-synuclein levels in wild type mice that don't over-express human mutant synuclein?

The PFF did not appear to produce robust formation of alpha-synuclein inclusions. There are published methods in the literature on how to optimize the PFF model which the authors could explore for future studies in order to obtain robust dopamine neuron loss using this model. In the future it would be nice to see PFF injections first, followed by treatment with TAT-betasyn-degron to determine if the progression of Lewy-body-like aggregates can be stopped.

For Figure 4 and 6- I like these experiments because knocking out alpha-synuclein protects dopamine neurons from MPP+. Thus the data here support that reducing synuclein can be protective against MPP+ induced toxicity.

In figure 7, it is interesting that the Tat-betasyn peptide was protective for the rotarod test but did not protect against MPTP-induced TH neuron cell loss or reduce synuclein levels. In addition to the possible inhibition of synuclein oligomerization in dopamine neurons, it seems that the TAT-betasyn peptide was somewhat protective against dopamine terminal loss in the striatum. Do the authors think this could contribute to the protective effect of the TAT-betasyn peptide?

Could the authors discuss the feasibility of using this technique in humans? Are there any preclinical trials using the TAT-degrons for any protein? Would there be any side effects or off target effects?

Minor comments:

In figure 4, please include representative images of the immunocytochemistry staining. '

Implement supplemental data 3 into page 10 of results, not in the discussion, perhaps line 226. Edit accordingly in discussion.

In figure 4, the order of the graphs (from left to right) is control, MPP, Tat- β syn, and Tat- β syn-degron. Keep same order consistent with Figures 6 and 7.

In line 182, include the mentioned behavioral data of rotarod test in supplemental data.

Point-by-point response

Reviewers' comments:

Reviewer #1 (Remarks to the Author):

In this manuscript, Jin, Fan and Cid-Pellitero with colleagues describe the modification of a previously developed (Fan et al., Nat Neurosci. 2014) blood-brain barrier and plasma membrane-permeable α -synuclein knockdown peptide, Tat- β syn-degron. They describe how the peptide is able to reduce a-syn levels via proteasomal degradation in cell and animal models. Specifically, the peptide is able to reduce a-syn levels and microglial activation in a pre-formed fibril model of a-syn propagation in human A53T overexpressing mice. In a MPTP -based PD model, the peptides reduce motor dysfunction. Overall, this is an interesting paper that contributes to furthering the knowledge of targeting a-syn to the field. However, I have some comments regarding different aspects of the manuscript that need to be addressed.

1. How do you know the reduction peaks at 24h (Fig 3B) when no timepoint after 24h is shown here? In vivo data from M83 line confirms this statement, but no such conclusions can be drawn from 3B alone.

Response: We are sorry for the inaccurate description. We fully agree with the reviewer and have now rephrased the sentence in the manuscript as “Additionally, the knockdown is time dependent, and the α -synuclein level remained low at 24 hrs (**Fig. 3b**).” The change is highlighted in red in the manuscript.

2. Fig 5, is it possible to add a visual to clarify experimental timeline? Text from the result section and the methods do not seem to match.

Line 173-176: Starting three days prior to the PFF injection, mice were treated daily for 9 days with either Tat- β syn or Tat- β syn-degron peptide (40 mg/kg; i.p.). Subsequently, the mice were treated with the same dose every other day for an additional 8 days.”

Line 500-503: Mice received intraperitoneal (i.p) injections of PBS, Tat-βsyn-degron peptide (40 mg/kg) or Tat-βsyn peptide (40 mg/kg), daily for 12 days, starting 3 days before the intracerebral (i.c.) injection, and subsequently every other day for 8 days (20 days in total).”

Response: We are sorry for the confusion. As suggested by the reviewer, we have added a timeline scheme in Figure 5a to clarify the experimental timelines. The text and figure legend have also been modified accordingly. All the changes are highlighted in red.

3. PFF model – the brain regions evaluated, how are they connected to site of injection (i.e., how many synapses apart) and each other?

Response: It is well known that intracerebral (i.c.) PFF injection is linked to α- synuclein spreading over multiple synapses in various brain areas through both synaptic and non-synaptic transmission, including diffusion through the cerebral spinal fluid (Luk et al, J Exp Med, 2012). We analyzed the nigrostriatal pathway (striatum and SN) and several other brain areas, such as hypothalamus, pons, and medulla oblongata. The primary aim of the experiment was to determine whether the Tat-βsyn-degron peptide could affect the propagation of α-synuclein. We found statistical differences between the treated group and the control group in the SN and the pons. The precise anatomical pathways and the number of synapses in the pathways are beyond the scope of the present study, but will be important focuses of our future studies.

4. Why different treatment regime for the MPTP model vs the PFF model? Injected with the drug at different concentration (6umol/kg MPTP and 40mg/kg PFF) and twice per day for the MPTP model.

Response: The molecular weight of the Tat-βsyn-degron peptide is 3147 g/mol. 6 umol/kg Tat-βsyn-degron equals to 18.9 mg/kg. In the MPTP model, mice received two i.p. injections of Tat-βsyn-degron per day (18.9 mg/kg *2 = 37.8 mg/kg). In the PFF model, mice received one Tat-βsyn-degron peptide injection per day (40 mg/kg; i.p.).

The PFF model was conducted at our collaborator's lab at McGill University, Canada and the MPTP model was conducted at another collaborator's lab at Chongqing Medical University, China. There was some inconsistency in experiment design when the experiments were conducted at two different labs and we are sorry for this. However, as you can see from the calculation above, mice from both models received roughly the same total amount of the Tat- β syn-degron peptide per day.

5. Fig 6 – As I interpret it, all significance shown is referring to significant differences between the control group and, like in 6B, the MPTP/Tat- β syn-degron group. I think this could be clarified in the main text, only the last sentence in the figure legend explains that everything is rereferring back to the control. I.e., in none of the graphs are there any comparisons being made between the MPTP group and the MPTP/peptide group.

Response: sorry for the confusion and thanks for the suggestion. Comparisons between the MPTP group and the MPTP + peptide group have been made and indicated in figures 4, 6 and 7 and corresponding figure legends. It has been made clear in all the figure legends that * $P \leq 0.05$, ** $P \leq 0.01$ and *** $P \leq 0.001$ refer to comparison with the control; # $P \leq 0.05$, ## $P \leq 0.01$ and ### $P \leq 0.001$ indicate the statistical difference between the MPTP + peptide group and the MPTP group. n.s. denotes not significant.

Confusing to reader when Fig 7E is referred to in the figure legend for Fig 6.

Response: We are sorry for the confusion. We have removed “see **Fig. 7e**” in Fig. 6's figure legend.

6. How is lowering a-syn levels protecting dopaminergic neurons in the MPTP model; it's not known to robustly induce a-syn pathology so what is the mechanism of action here? The authors mention a-syn oligomerization in the discussion, so it would be very interesting to look at oligomerized synuclein to further 1) justify using the MPTP model for evaluating a a-syn targeting drug candidate and 2) characterize the peptides effect (both Tat- β syn or Tat- β syn-degron)?

Response: We thank the reviewer for the suggestion. The role of α -synuclein knockdown in preventing MPP⁺/MPTP-induced neuronal toxicity has been explored and validated in several previous studies. One study showed that α -synuclein null mice display striking resistance to MPTP-induced degeneration of dopaminergic neurons and dopamine release, and this resistance appears to result from an inability of the toxin to inhibit mitochondrial complex I (Dauer et al., Proc Natl Acad Sci, 2002). Another study showed that RNAi-mediated α -synuclein knockdown prevents MPP⁺-induced activation of nitric oxide synthase (NOS) in models of human neurons (Fountaine et al., Eur J Neurosci, 2008). α -synuclein aggregation, a pathological hallmark of PD, is not commonly observed in the MPTP mouse model. However, under certain conditions, chronic MPTP infusions can induce α -synuclein oligomerization in mice (Fornai et al., Proc Natl Acad Sci, 2005; Shioda et al., JBC, 2014; Matsuo et al, Neuropharmacology, 2019). We agree with the reviewer that it would be interesting to investigate α -synuclein oligomerization in our MPTP model. However, as correctly noted by the reviewer, it's not known that the MPTP model can robustly induce α -synuclein pathology. Therefore, it may not be a proper model for such a study even if the Tat- β syn-degron peptide did protect neurons through reducing the non-detectable α -synuclein oligomerization. Due to this reason, we have supplemented this MPTP model with a PFF model of PD. Given the clear function of α -synuclein knockdown in reducing neuronal damage by our Tat- β syn-degron peptide in two different animal models of PD in this study, we respectfully argue that our results are sufficient to support the conclusion of this study that the α -synuclein knockdown peptide has therapeutic potential for PD. We can further study the detailed mechanisms by which the peptide exert its therapeutic effects in our future study.

7. In the methods it is stated that the TH neurons were manually counted in three nigral sections within a set distance. This would preferably be done using stereology to recapture the most accurate, unbiased cell count.

In addition, the methods describe that the number of cells counted in these three nigral sections were multiplied by 6. I assume this is because only 1/6 of the brain was stained.

However, the distribution of cells is not the same throughout the entire nigra, thus multiplying by 6 will by no means give an accurate estimate of the total number of cells.

In addition, why was only three nigral sections counted? In a 6-series of a mouse brain sectioned in 30um intervals, around 6-7 sections should contain the nigra, counting all of those would give a more accurate estimate.

Response: We agree with the reviewer that quantitatively counting the TH-positive neurons in SNpc using stereology is important. We are sorry to say that we do not have access to the software (such as Stereo Investigator, MicroBrightField) and the equipment for unbiased stereology. Alternatively, we sliced the mouse substantia nigra into a series of 30 μ m sections (from rostral to caudal), and stained one in every six sections with anti-TH antibody. The substantia nigra pars compacta was outlined using TH-positive neurons between identifiable landmarks from bregma -2.92mm to -3.64mm. Identifiable TH-positive neurons were manually counted by an experimenter blinded to the treatment conditions in three stained sections within this distance from each animal, and the total number of the TH-positive neurons in this region was estimated using the following equation: $N = \text{counted number} \times 6$, as described in the method section. We agree with the reviewer that blindly counting 3 stained sections from bregma -2.92mm to -3.64mm and multiplying the counted number by 6 do not represent the total number of the dopaminergic neurons within SNpc. However, since samples from all groups were processed equally, the counting results do semi-quantitatively estimate the dopaminergic neurons from bregma -2.92mm to -3.64mm in the substantia nigra pars compacta. That's why the counting results shown in our Fig. 7b are consistent with the TH immunoblotting results in the substantia nigra-containing ventral midbrain as shown in Fig. 6c. As per the reviewer's suggestion, we have changed the wording in the result section to "there was a significant loss of TH-positive neurons in the substantia nigra pars compacta as estimated by blinded neuron counting from bregma -2.92mm to -3.64mm (**Fig. 7a and 7b**)". A similar change in wording was also made in Figure 7's legend to better describe the blind-counting results. And Fig. 7b is now labelled as "Estimated numbers of TH-positive neurons". We hope all these changes will end the confusion of future readers.

8. For the statistical analysis, only for the rotarod data was analyzed using a two-way ANOVA (as is appropriate). The remained of the data was analyzed using a student's t-test (for aSyn data in fig 4, also appropriate since the PBS group can be excluded). However, one-way ANOVA and following post hoc tests were performed for the reminder of the data; only comparing the control group with the additional groups. In fig 6F and 7D, # is indicating a trend between the MPTP and the MPTP/ Tat- β syn-degron (not mentioned in the text nor the figure legend). But this cannot be accurately evaluated using a one-way ANOVA since there are two main effects evaluated; 1) if MPTP has an effect and 2) if the treatment with the peptide has an effect, which requires a two-way ANOVA to address.

Response: We thank the reviewer for the suggestion. We have re-analyzed the data in Fig. 4b-c, Fig. 4e, Fig. 6b-c, Fig. 6e-f, Fig. 7b, and Fig. 7d using two-way ANOVA and the analysis results are shown in the figure legends. The statistical difference in neuroprotection between the MPTP group and the MPTP + peptide group is analyzed by post hoc analysis and has been clarified in the figure legends. Although the method of analysis has been changed, the conclusion of our study remains the same.

Reviewer #2 (Remarks to the Author):

In this manuscript authors have evaluated the efficacy of a a-syn-binding synthetic peptide in mitigating the PD-like neuropathology in mice. Study has utilized both in vitro and in vivo approaches. This manuscript can be published with minor revisions.

1. Title, Summary and Discussion. Title of the manuscript is misleading. It should reflect the content of the manuscript. Authors show that their peptide in most effective in MPTP-Induce mice. They did not test A53T mice. Their experiment with a-syn spreading model was not successful. The abstract and discussion should also modify accordingly.

Response: we thank the reviewer for the suggestion. We have modified the title, the abstract and the discussion accordingly to better reflect the content of the manuscript. All the changes are highlighted in red.

2. Fig. 6. In this fig, authors show that b-syn-degron reduces a-syn and mitigates neuronal loss. This observation is consistent with previous reports that suggest that reducing a-syn is beneficial in PD-mouse models. However, this Fig. shows that MPTP does not alter the a-syn level in the brain, this appears to contradict previous reports.

Response: the increase of α -synuclein in the brain after the MPTP treatment seems to be time-dependent and region-specific in mice. A previous paper that used the same chronic MPTP injection protocol as ours (Vila et al., J Neurochem, 2000) showed that “after the chronic MPTP regimen (C57BL mice received one i.p. injection of MPTP - HCl per day (30 mg/kg) for 5 consecutive days and were sacrificed at 0, 2, 4, 7 days after the last injection for immunoblotting analysis), α -synuclein protein expression progressively increased in midbrain extracts from 0 to 4 days after MPTP administration. This increase was already significant at 0 days (+44%), peaked at 4 days (+77%), and then returned to control level at 7 days. No changes in α -synuclein protein levels were detected in the striatum or in other cerebral structures, such as cerebellum or cortex”. In our experiment, the C57BL/6 mice received i.p. injection of 30 mg/kg MPTP hydrochloride once a day for 5 days and were sacrificed for immunoblotting analysis 7 days after the last MPTP injection. Therefore, it is likely that MPTP-induced α -synuclein protein increase had returned to the baseline level on the seventh day after the last MPTP injection. Maybe that’s the reason why we did not see an increase of α -synuclein in both tissues (ventral midbrain and striatum) after the MPTP treatment in Fig. 6.

Reviewer #3 (Remarks to the Author):

This study focused on the development of an α -synuclein knockdown peptide, Tat- β syn-degron. By utilizing both in vitro and in vivo Parkinson’s disease (PD) models, they observed a decrease of α -synuclein in Tat- β syn-degron while decreasing neuronal damage and motor impairment. They authors concluded that this peptide could lead to a

therapeutic treatment in PD. This is a nice paper showing the potential for this technique to reduce alpha-synuclein and possibly protect against progression in PD. Also, I think this technique has a lot of potential as a tool in the lab for reducing protein expression or targeting specific domains within proteins.

1. In the cortical neurons and the mouse brain, are levels of beta-synuclein reduced? This may be difficult to show by Western blot if the authors cannot find a good beta-synuclein specific antibody. Could the authors use qPCR to show if beta-synuclein is reduced in the neurons with exposure to the TAT-betasyn-degrom? Examining gamma-synuclein is not necessary because it is not highly expressed in neurons.

Response: we agree with the reviewer that testing β -synuclein is necessary to investigate the specificity of the Tat- β syn-degrom peptide. As mentioned by the reviewer, we could not find a good, clean anti- β -synuclein antibody. Since our knockdown peptide reduces synuclein at the protein level, not at the mRNA level, qPCR won't tell whether the β -synuclein protein level is reduced or not. Alternatively, as shown in Fig. 1f, we co-transfected HEK cells with α -synuclein, HA- β -synuclein or HA- γ -synuclein plasmid together with the plasmid encoding the knockdown peptide or the control peptide. We were able to demonstrate that the knockdown peptide can only reduce the level of α -synuclein, but not HA- β -synuclein nor HA- γ -synuclein. These results suggest that this α -synuclein knockdown peptide specifically targets α -synuclein.

2. I am unclear why the authors used beta-synuclein as targeting sequence besides that a previous study showed this sequence binds to alpha-synuclein. It makes sense as a proof-of-principle that this technique can work. However, I would like to see a targeting sequence specific for alpha-synuclein such as starting around amino acid 72- for future studies at least.

Response: we thank the reviewer for the great suggestion. In the current study, we used the β -synuclein sequence as it binds to α -synuclein with high affinity and can be a good choice for our proof-of-principle study. We definitely agree with the reviewer and will try to develop a next generation α -synuclein knockdown peptide with a more specific

binding sequence for α -synuclein in our future studies.

3. It looks like in primary neurons, there is a reduction in beta-actin with higher concentrations or longer time points of exposure with the TAT-betasyn-degrom. It would be nice to see a toxicity assay with the TAT-betasyn-degrom compared to the control peptide and no treatment.

Response: following the reviewer's suggestion, we treated primary cortical neurons with different doses of the Tat- β syn and Tat- β syn-degrom peptides and then harvested the culture medium 24 hours later to measure the cytotoxicity. As shown in our new Supplementary Fig. 1, both the Tat- β syn and Tat- β syn-degrom peptides induced similar and dose-dependent increases of LDH release in the primary cortical neurons. We think this non-selective cytotoxicity is due to the pore-forming property of the Tat peptide on the cell plasma membrane (Singh et al., Drug Deliv, 2018). This phenomenon may only exist in culture cells and does not incur any safety concern *in vivo* because the Tat peptide is safe in humans and a Tat-fused peptide has even successfully passed a phase 3 clinical trial recently (Hill et al., Lancet, 2020).

4. For figure 3 and figure5- did the authors use a human specific synuclein antibody to confirm reduced levels of alpha-synuclein with TAT-betasyn-degrom knockdown? Or did they use an antibody that recognizes both mouse and human synuclein? Did the authors perform western blots from forebrain homogenates from these mice? Did the TAT-betasyn-degrom reduce alpha-synuclein levels in wild type mice that don't over-express human mutant synuclein?

Response: In Fig 3f, we used an ELISA kit (KHB0061, ThermoFisher Scientific) that is specific for human α -synuclein. In Fig 5, the synuclein antibody (ab184674) used in the study recognizes the phosphorylated Serine129 (pS129) of the mouse and human α -synuclein, which is a marker of pathogenic synuclein aggregates (Oueslati, J Parkinsons Dis, 2016). In Fig. 3f and Fig. 5, all our studies were performed using the M83 mouse line, which overexpress human A53T mutant α -synuclein. Western blots were not performed in these M83 mice. The Tat- β syn-degrom peptide could reduce α -

synuclein in the wild type mice and the results are shown in Fig. 6 and Supplementary Fig. 5.

5. The PFF did not appear to produce robust formation of alpha-synuclein inclusions. There are published methods in the literature on how to optimize the PFF model which the authors could explore for future studies in order to obtain robust dopamine neuron loss using this model. In the future it would be nice to see PFF injections first, followed by treatment with TAT-beta-syn-degron to determine if the progression of Lewy-body-like aggregates can be stopped.

Response: We thank the reviewer for these excellent suggestions and we will perform these suggested investigations in our future work.

6. For Figure 4 and 6- I like these experiments because knocking out alpha-synuclein protects dopamine neurons from MPP+. Thus the data here support that reducing synuclein can be protective against MPP+ induced toxicity.

Response: Thank you.

7. In figure 7, it is interesting that the Tat-beta-syn peptide was protective for the rotarod test but did not protect against MPTP-induced TH neuron cell loss or reduce synuclein levels. In addition to the possible inhibition of synuclein oligomerization in dopamine neurons, it seems that the TAT-beta-syn peptide was somewhat protective against dopamine terminal loss in the striatum. Do the authors think this could contribute to the protective effect of the TAT-beta-syn peptide?

Response: we agree with the reviewer that the protective effects of Tat- β syn peptide are interesting. In Fig. 7c-d, the Tat- β syn peptide did induce a mild protection against MPTP-induced loss of dopaminergic neuron terminals in the striatum. We agree with the reviewer that this may partially contribute to the protective effects of the Tat- β syn peptide because the signalling pathway from the substantia nigra to the striatum is partially protected.

8. Could the authors discuss the feasibility of using this technique in humans? Are there any preclinical trials using the TAT-degrons for any protein? Would there be any side effects or off target effects?

Response: we thank the reviewer for the great suggestion. A recent success in a phase 3 clinical trial has already demonstrated that a Tat-fused short peptide is not only safe, but also therapeutically effective in protecting neurons against ischemic damage in humans (Hill et al., Lancet, 2020). Based on this promising human clinical trial, we feel strongly that our protein knockdown technique may be clinically applicable for human uses. As far as we know, we are the first ones to use TAT-degron peptides for protein knockdown. Admittedly, as a proof-of-principle study, we have not carefully ruled out off-target effects of the peptide. But, through further testing and optimization, we think this α -synuclein knockdown peptide may be developed into a new, effective, and clinically practical therapeutic treatment for PD in human patients. We have added these information in the discussion section of the revised manuscript. All the changes are highlighted in red.

Minor comments:

1. In figure 4, please include representative images of the immunocytochemistry staining.

Response: we thank the reviewer for the suggestion. We have added the representative images in the new Fig. 4d.

2. Implement supplemental data 3 into page 10 of results, not in the discussion, perhaps line 226. Edit accordingly in discussion.

Response: we thank the reviewer for the suggestion. We have moved the supplementary figure 3 (Now renamed as supplementary Fig. 5) to the result section and made the edits accordingly in the discussion section. All the changes are highlighted in red.

3. In figure 4, the order of the graphs (from left to right) is control, MPP, Tat- β syn, and

Tat- β syn-degron. Keep same order consistent with Figures 6 and 7.

Response: Fig. 4 was produced at University of British Columbia, Canada and Fig. 6 and 7 were produced in Chongqing Medical University, China. We are sorry for the inconsistency in figure formatting. It has been a while since we did the experiments in Fig. 6 and 7 and mouse tissue samples may not be available or fresh enough to re-do the immunoblotting in a new format. We do thank the reviewer for the great suggestion and we will keep this in mind in our future studies.

4. In line 182, include the mentioned behavioral data of rotarod test in supplemental data.

Response: we thank the reviewer for the suggestion. We have added the behavioral data as supplementary Fig. 3.

REVIEWERS' COMMENTS:

Reviewer #2 (Remarks to the Author):

Authors have addressed my concerns.

Reviewer #3 (Remarks to the Author):

The authors did a very nice job responding to the reviewers comments.

Author responses

REVIEWERS' COMMENTS:

Reviewer #2 (Remarks to the Author):

Authors have addressed my concerns.

Author response: thank you!

Reviewer #3 (Remarks to the Author):

The authors did a very nice job responding to the reviewers comments.

Author response: thank you!